# Programmed cell removal by calreticulin in tissue homeostasis and cancer

Mingye Feng[1,2], Kristopher D. Marjon[1,3,4], Fangfang Zhu[1,3,4], Rachel Weissman-Tsukamoto[1,3,4], Aaron Levett[1,3,4], Katie Sullivan[1,3,4], Kevin S. Kao[1,3,4], Maxim Markovic[1,3,4], Paul A. Bump[1,3,4], Hannah M. Jackson[1,3,4], Timothy S. Choi [1,3,4], Jing Chen[2,5], Allison M. Banuelos[1,3,4], Jie Liu[1,3,4], Phung Gip[1,3,4], Lei Cheng[5], Denong Wang[6] & Irving L. Weissman [1,3,4,7]

Macrophage-mediated programmed cell removal (PrCR) is a process essential for the clearance of unwanted (damaged, dysfunctional, aged, or harmful) cells. The detection and recognition of appropriate target cells by macrophages is a critical step for successful PrCR, but its molecular mechanisms have not been delineated. Here using the models of tissue turnover, cancer immunosurveillance, and hematopoietic stem cells, we show that unwanted cells such as aging neutrophils and living cancer cells are susceptible to "labeling" by secreted calreticulin (CRT) from macrophages, enabling their clearance through PrCR. Importantly, we identified asialoglycans on the target cells to which CRT binds to regulate PrCR, and the availability of such CRT-binding sites on cancer cells correlated with the prognosis of patients in various malignancies. Our study reveals a general mechanism of target cell recognition by macrophages, which is the key for the removal of unwanted cells by PrCR in physiological and pathophysiological processes.

[1] Institute for Stem Cell Biology and Regenerative Medicine, Stanford University, Stanford, CA 94305, USA. [2] Department of Immuno-Oncology, Beckman Research Institute, City of Hope Comprehensive Cancer Center, Duarte, CA 91010, USA. [3] Ludwig Center for Cancer Stem Cell Research and Medicine, Stanford University, Stanford, CA 94305, USA. [4] Stanford Cancer Institute, Stanford University, Stanford, CA 94305, USA. [5] State Key Laboratory of Oral Diseases and National Clinical Research Center for Oral Diseases and West China Hospital of Stomatology, Sichuan University, Chengdu, Sichuan 610041, China. [6] SRI International, Menlo Park, CA 94025, USA. [7] Department of Pathology, Stanford University, Stanford, CA 94305, USA. Correspondence and requests for materials should be addressed to M.F. (email: mfeng@coh.org) or to I.L.W. (email: irv@stanford.edu)

The process of viable cell clearance via phagocytosis by macrophages was termed by us as "programmed cell removal" (PrCR), which consists of multiple steps, including recognition, cellular engulfment, and intracellular digestion of the target cells[1–4]; and is conserved in many metazoan species[5–8]. The phagocytosis of cells undergoing PCD but that have not yet burst presumably prevents dying cell contents from causing inflammation[4,9]. While phagocytosis is often tied to programmed cell death (PCD) as it is essential for the clearance of apoptotic cells, called efferocytosis[9], PrCR can occur in many circumstances independently of PCD[1,3,4,10,11]. We have shown that while the initiation of both PCD and PrCR can occur in aging neutrophils, when PCD is blocked by enforced expression of Bcl2, PrCR is not blocked and results in physiological removal of the neutrophils[3]. PrCR of living cells plays integral roles in many physiological and pathophysiological processes, including inflammation, hematopoiesis, tissue turnover, and cancer immunosurveillance[1,3,12]. During these process, viable target cells are cleared by macrophages in PrCR without PCD being induced. The efficacy of PrCR is determined by the balance between the recognition of pro-phagocytic "eat me" signals by macrophages and the inhibition of macrophages via the activation of anti-phagocytic "don't eat me" pathways by target cells. Cancer cells that have upregulated the "don't eat me" signal CD47 inhibit PrCR by macrophages through signaling via macrophage signal regulatory protein α (SIRPα)[4,13,14]. Blockade of CD47 on cancer cells leads to their recognition and phagocytosis via a cell surface form of the effector calreticulin (CRT) on macrophages[1,2,15–17]. The binding of the cancer cell's CD47 to the macrophage's SIRPα receptor leads to SHP-1 and/or SHP-2 activation. These tyrosine phosphatases inhibit phagocytosis of macrophage-bound targets, at least in part by dephosphorylation of the actin-myosin-paxillin components required for engulfment and phagocytosis[18]. In the ER lumen, CRT functions as a chaperone to assist folding and assembly of a list of cell surface and secreted proteins, including major histocompatibility complex (MHC) class I[19], as well as ER resident proteins including other chaperones. CRT has also been found to regulate adhesion through integrin activation[20] and integrins have been demonstrated to regulate CRT presentation on the cell surface[21]. Interaction between Thrombospondin-1 (TSP1)[22] and CRT on the cell surface has been implicated to signal through low-density lipoprotein (LDL) receptor-related protein (LRP1 or CD91) to induce focal adhesion disassembly. On the cell surface, CRT, C1q and CD91 receptors can form bridging complexes between macrophages and apoptotic cells, activating cellular machinery responsible for initiating phagocytosis of apoptotic cells[10,23–25]. Most CD47+ normal tissue cells do not become susceptible to PrCR when their CD47 signal is blocked, due to the lack of "eat me" signals that are necessary for macrophage recognition[1,14]. Thus the macrophage integrates pro- and anti-phagocytic signals for each cellular target that is then either phagocytosed or allowed to remain. Such a system of regulated opposing signals stringently defines the specificity and selectivity of PrCR for the clearance of unwanted cells (Supplementary Fig. 1a). It enables the induction of PrCR as a promising cancer treatment approach with high efficacy and minor non-specific toxicity.

Despite its importance, little is known regarding how macrophages detect and recognize target cells during PrCR and whether a general mechanism is shared among PrCR of different types of unwanted cells including damaged, dysfunctional, aged, and malignant cells. While some similarities have been identified between the PrCR of apoptotic cells and that of living cells, there is emerging evidence that distinct mechanisms may regulate these two processes[4]. In our recent studies, we identified CRT as a "guide" for targeting living cancer cells by macrophages[2]. We showed that CRT moves from the endoplasmic reticulum of macrophages to the cell surface and/or is secreted following macrophage activation via toll like receptors[2], a process that involves BTK phosphorylation of endoplasmic reticulum CRT, cleavage to separate the ER retention KDEL from the rest of the protein, and secretion to bridge macrophage CD91 prophagocytic receptors to appropriate target cells[23–26].

Here we dissect the mechanism of CRT-mediated recognition of living cells destined for PrCR. Our study demonstrates that intrinsic changes in target cells result in their susceptibility to bind macrophage-secreted CRT, which labels the target cells for subsequent removal by PrCR. We reveal a surprising finding—the cells which are to undergo PrCR reveal a CRT-binding moiety on their cell surface, and the moiety is the family of asialoglycans. The expression of pathways regulating CRT binding in human cancers is correlated with the prognostic outcomes of patients with those cancers. The generation of CRT-binding sites on cancer cells can therefore enhance PrCR and could be important in developing highly efficient cancer treatment strategies. We show that this is a general mechanism of the interaction between macrophages and their target cells during PrCR in many physiological and pathophysiological conditions. In at least one case, the target cell induces cell-surface asialoglycoproteins by the function of neuraminidase 4. The sharing of this mechanism with PrCR of non-malignant cells opens the path to testing therapies for disease cells protected by CD47, including a wide range of human cancer cells, atherosclerotic cells, and fibrotic disease cells[27,28].

## Results

**Aged neutrophils are cleared through activation of PrCR.** During sterile or infection-induced peritonitis, immature neutrophils leave the bone marrow, infiltrate the inflammation site and mature there[29,30]. These neutrophils are programmed to initiate age-dependent cell apoptotic processes, but to be phagocytosed by peritoneal macrophages before the explosive release of cellular contents that would cause inflammation; they burst inside of macrophages. We utilized a mouse model in which neutrophils are resistant to cell death by enforced expression of Bcl2 under the promoter MRP8[3,31]. In agreement with our previous findings using this animal model, we found after thioglycollate induced peritonitis, the Bcl2-expressing neutrophils were resistant to apoptosis but were removed from the peritoneum with the same temporal dynamics as WT neutrophils undergoing cell death (Fig. 1a, b and Supplementary Fig. 1b). These data indicate that aged neutrophils are programmed to be removed through PrCR; therefore, recognition of such target cells by macrophages is independent of their cell death. We reasoned that both cell death and PrCR programs were activated in wild-type (WT) neutrophils to promote their removal in the peritoneum, while only PrCR was activated in the Bcl2-protected neutrophils (Fig. 1c). To reveal potential pathways utilized by neutrophils to activate PrCR, we performed gene-expression profiling using RNAseq analysis to compare immature bone-marrow neutrophils to mature peritoneum-infiltrating neutrophils from WT and *MRP8-Bcl2* mice (Fig. 1d and Supplementary Fig. 1b) at the time the neutrophils are starting to be removed from the peritoneum. We identified 333 overlapping genes that are upregulated in both WT and Bcl-2-expressing neutrophils that are potentially involved in PrCR (Fig. 1d and Supplementary Fig. 1c). Gene ontology enrichment pathway analyses revealed enriched pathways for MAPK signaling cascade, response to stress, and immune system processes (Fig. 1e).

**Cell surface CRT determines PrCR of viable aged neutrophils**. We wondered whether modifications on the cell surface were changing to promote cell communication leading to PrCR. We evaluated the cell surface changes on neutrophils and macrophages for "eat me" and "don't eat me" signals, such as CRT and CD47, respectively, that regulate PrCR after thioglycollate injection into *MRP8-Bcl2* mice (Fig. 2a, b). We found that PrCR of neutrophils after thioglycolate injection correlated

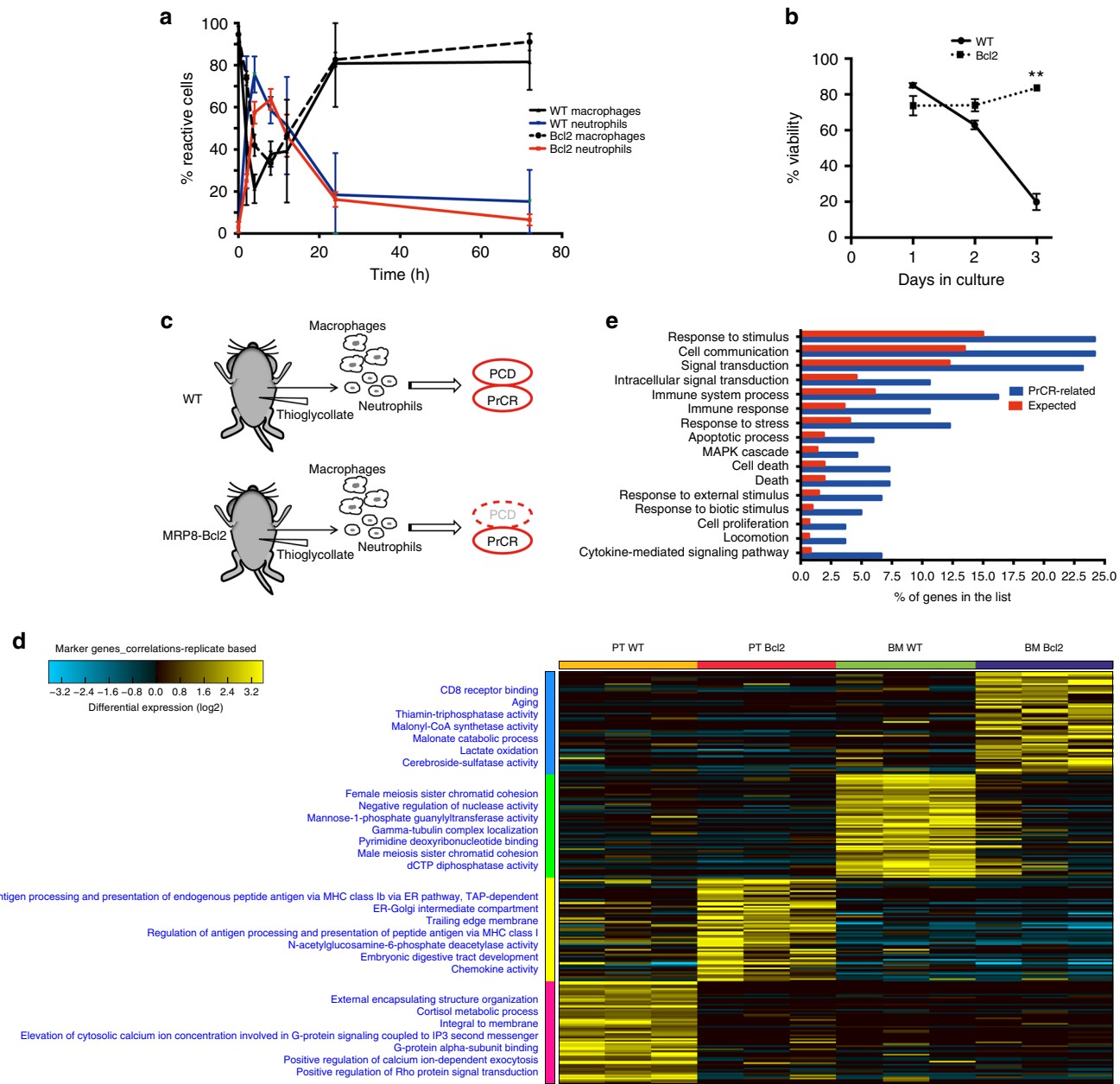

**Fig. 1** Viable aged neutrophils are cleared through activation of program cell removal. **a** Analysis of neutrophil and macrophage cell populations after thioglycollate injection to WT and *MRP8-Bcl2* mice. Peritoneal cells were collected and analyzed at 0, 2, 4, 8, 12, 24, and 72 h after thioglycollate injection. Similar lifespan was observed for WT and Bcl2 macrophages and neutrophils. Macs, macrophage;s WT, wild-type; PMN, neutrophil/polymorphonuclear cell. Results are representatives of three independent experiments. $n = 3$ mice for each time point. Error bars represent standard deviation. Cells were identified by surface markers such as CD11b$^+$F4/80$^-$GR1$^{++}$ (Neutrophils), CD11b$^+$F4/80$^+$ (macrophages). Percentage reactive cells, the % of the total viable cells found in the peritoneum after induction of peritonitis. **b** WT but not Bcl2 neutrophils undergo cell death. WT and Bcl2 neutrophils were collected and cultured in vitro for 72 h. Cell viability was examined by AnnexinV and DAPI staining. Cells that were AnnexinV-DAPI- were considered as viable cells. $n = 3$. **$P < 0.01$ (t-test) for viability between WT and Bcl2 neutrophils. Error bars represent standard deviation. **c** A schematic showing peritonitis in WT and *MRP8-Bcl2* mice. WT neutrophils undergo both programmed cell death (PCD) and programmed cell removal (PrCR) after maturation while Bcl2 neutrophils are resistant to PCD but maintain PrCR programs. **d** RNAseq anaylsis of 4 groups of neutrophils, including (1) WT neutrophils from bone marrow (BM), (2) WT neutrophils recruited to peritoneum (PT), (3) *MRP8-Bcl2* neutrophils from bone marrow, and (4) *MRP8-Bcl2* neutrophils recruited to peritoneum. PT, peritoneal; BM, bone marrow. **e** Distribution among different cellular pathways (y-axis) of: (blue) 333 genes that are associated with increased susceptibility to PrCR and (red) genes in the human genome. RNAseq analysis was performed on 4 groups of neutrophils, as described in **d**. RNAseq analysis was used to identify genes upregulated in the maturation process (BM to peritoneum) in both WT and *MRP8-Bcl2* mice, which are likely associated with susceptibility to PrCR

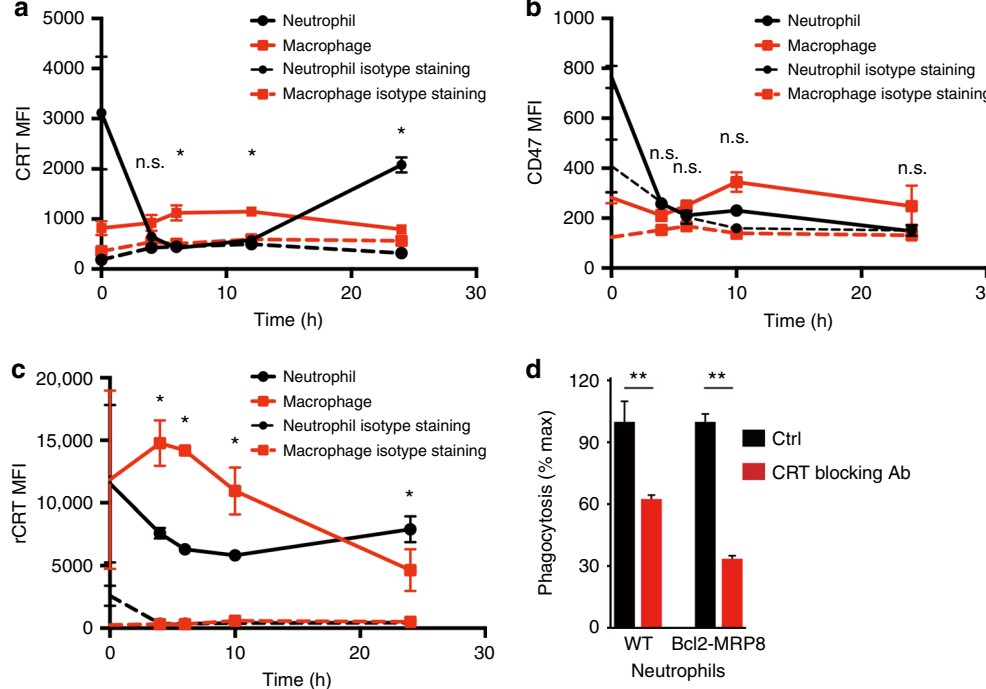

**Fig. 2** Cell surface CRT determines programmed cell removal of neutrophils in peritonitis. **a**, **b** Cell surface levels of CRT on macrophages and neutrophils after thioglycollate injection to *MRP8-Bcl2* mice. Peritoneal cells were collected and analyzed at 0, 4, 6, 12, and 24 h after thioglycollate injection, by flow cytometry analysis. Results are representatives of three independent experiments. $n = 3$ mice for each time points. *$P < 0.05$ (paired $t$-test) for CRT staining between macrophages and neutrophils at the same time points. Error bars represent standard deviation. **b** Cell surface levels of CD47 on macrophages and neutrophils after thioglycollate injection to *MRP8-Bcl2* mice. Peritoneal cells were collected and analyzed at 0, 4, 6, 10, and 24 h after thioglycollate injection, by flow cytometry analysis. Results are representatives of three independent experiments. $n = 3$ mice for each time point. n.s. (paired $t$-test) so significant differences for CD47 staining between macrophages and neutrophils at the same time points. Error bars represent standard deviation. **c** Binding of recombinant CRT to macrophages and neutrophils after thioglycollate injection to *MRP8-Bcl2* mice. Peritoneal cells were collected and analyzed at 0, 4, 6, 10, and 24 h after thioglycollate injection. CRT-binding sites on macrophages and neutrophils were measured by incubating the cells with saturation concentration of recombinant CRT and analyzing by flow cytometry. Results are representatives of three independent experiments. $n = 3$ mice for each time point. *$P < 0.05$ (paired $t$-test) for rCRT staining between macrophages and neutrophils at the same time points. Error bars represent standard deviation. **d** An in vitro phagocytosis assay showing blockade of CRT inhibits phagocytosis of neutrophils, with WT and Bcl2 neutrophils as target cells and peritoneal macrophages. Neutrophils were collected 4 h after thioglycollate treatment and cultured for 24 h. Phagocytosis was normalized to the maximal response in the experiments. $n = 3$. **$P < 0.01$ ($t$-test) for phagocytosis between ctrl and CRT blocking Ab treatment. Error bars represent standard deviation. In **a**, **b**, and **c**, MFI, mean fluorescence intensity

with their cell surface expression of CRT, but not CD47 (Fig. 2a and Supplementary Fig. 2a). Interestingly, the cell surface levels of CRT increased on the neutrophils over time which correlated with the loss of neutrophils in the peritoneum. These data indicated that the cell surface was dynamically changing over time for PrCR to occur by promoting CRT accumulation on the neutrophil. To test this we isolated macrophages and neutrophils after thioglycollate injection over a period of time and determined the capacity of the cells to bind recombinant CRT. Both macrophages and neutrophils over time modified the cell surface to become more or less receptive to bind exogenous CRT (Fig. 2c). As the macrophages became less receptive to bind CRT neutrophils became more receptive to bind CRT. These data indicated that cell surface CRT regulated PrCR of neutrophils in this model. To test this we conducted a phagocytosis assay in which we blocked CRT recognition with a CRT blocking antibody and found that the phagocytosis of viable neutrophils by macrophages under this condition was significantly decreased (Fig. 2d). These data emphasize that PrCR is a dynamic process and PrCR of viable cells is dependent largely on the appearance of cell surface targets for CRT binding.

**Macrophages produce and secrete CRT to label cells for PrCR.** We investigated the molecular mechanisms of CRT-mediated PrCR of neutrophils by examining the origin of cell surface CRT on deathless neutrophils. Interestingly, neutrophils express extremely low levels of CRT while macrophages express high levels of CRT at both the RNA and protein levels, suggesting that surface CRT molecules on neutrophils may originate from macrophages (Fig. 3a–c, Supplementary Fig. 2b). To test this we isolated peritoneal macrophages after thioglycollate injection and found that they readily secreted CRT (Fig. 3d). Stimulation of macrophages with LPS promoted CRT secretion, suggesting that macrophages utilize this mechanism of CRT secretion to promote PrCR (Supplementary Fig. 2c, d). To further evaluate the source of CRT we cultured neutrophils or macrophages alone or together and measured cell surface CRT levels. Neutrophils did not alter the cell surface levels of CRT with respect to time when they were cultured alone (Supplementary Fig. 2e), however, co-culture of neutrophils and macrophages separated by a 0.4-μm pore membrane in a Boyden chamber showed a rapid and significant increase of cell surface CRT on neutrophils (Fig. 3e and Supplementary Fig. 2f). To determine the possibility that macrophages could produce this CRT and transfer it to neutrophils, we

metabolically labeled activated macrophages with AHA (ʟ-azi-dohomoalanine), a methionine analog, to label all newly synthesized proteins and then co-cultured these cells with neutrophils separated from the macrophages by a 0.4-μm pore

membrane in a Boyden chamber (Fig. 3f). Through a "click" reaction we were able to incorporate biotin into AHA positive protein on neutrophils. Immunoprecipitating CRT followed by western blot analysis of the immunoprecipitate and probing for

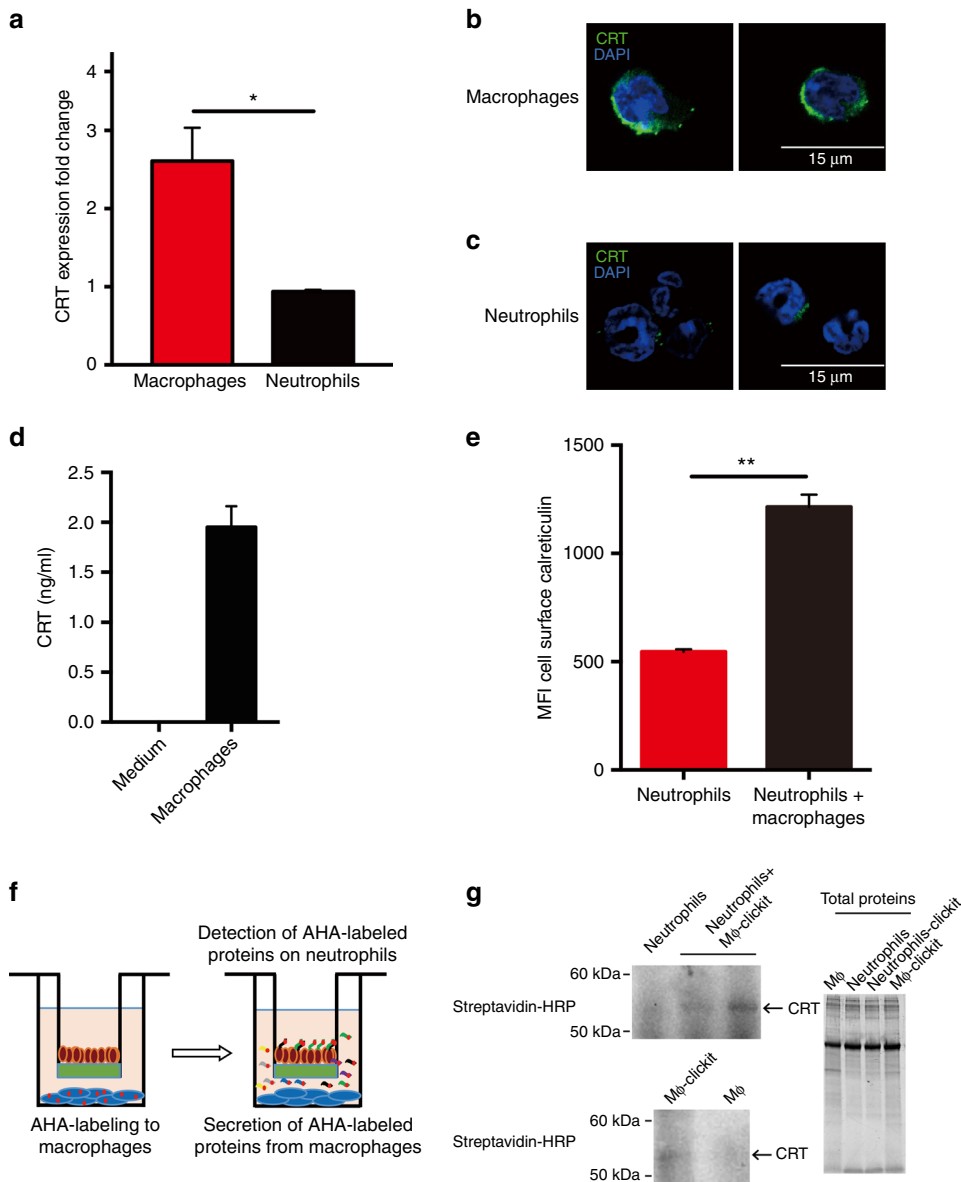

**Fig. 3** Macrophages are the sources of CRT and secrete CRT to label the target cells. **a** Expression of CRT measured by qRT-PCR in neutrophils and macrophages 8 h after thioglycollate. Macrophages and neutrophils were collected from *MRP8-Bcl2* mice as described in "Experimental Procedures". CRT mRNA level in macrophages were dramatically higher in macrophages as compared to neutrophils. *$P < 0.05$ (*t*-test) for expression of CRT between macrophages and neutrophils. Error bars represent standard deviation. **b**, **c** Immunofluorescent staining of CRT in mouse macrophages (**b**) and neutrophils (**c**). CRT is undetectable in neutrophils but abundant in macrophages. CRT localized to perinuclear regions, vesicles and cell surface of macrophages. Macrophages and neutrophils were collected from *MRP8-Bcl2* mice. **d** ELISA assay showing the amount of CRT in medium (RPMI) with and without macrophages. $n = 3$. Error bars represent standard deviation. Macrophages were able to secrete CRT to the extracellular medium. **e** Expression levels of cell surface CRT on neutrophils cultured alone or with macrophages in a 0.4-μm Boyden chamber overnight, assayed by flow cytometry. Co-culture with macrophages led to a significant increase of cell surface CRT on neutrophils. $n = 3$. **$P < 0.01$ (*t*-test) for expression levels of cell surface CRT on neutrophils cultured alone or with macrophages. Error bars represent standard deviation. **f** A schematic showing the Click-iT assay to examine transfer of CRT from macrophages to neutrophils during co-culture. Macrophages proteins were labeled with a methionine analog AHA (ʟ-azidohomoalanine) receptive for click chemistry. The proteins secreted by macrophages were all labeled with AHA. Because neutrophils and macrophages were cultured in a way independent of contact, the AHA-labeled proteins detected on neutrophils originate from the macrophages. **g** Neutrophils were cultured alone or together with AHA-treated macrophages in a 0.4-μm Boyden chamber. Cells were collected and CRT was immunoprecipitated and then subjected to click-chemistry adding in biotin to receptive methionine analogs. Samples were subjected to SDS-PAGE and then western blot for the presence of biotin. Detection of AHA-labeled CRT on neutrophils indicated that macrophage-secreted CRT to label neutrophils and this was independent of the contact between these two cell types

biotin with streptavidin-HRP, we found biotin positive CRT on neutrophils, indicating that macrophages secreted CRT that decorated the surface of the deathless neutrophils, in the absence of contact between the macrophages and neutrophils (Fig. 3g and Supplementary Fig. 2g). Considering the possibilities that macrophages modify target cells through cytokine secretion, we examined the cytokine profiling of macrophages under different circumstances. Blocking CRT from macrophages didnot affect their cytokine profiling, except for an upregulation of MIP2 and GROA, two inflammatory cytokines (Supplementary Fig. 2h). In addition, no significant differences were observed regarding the cytokine secretion from macrophages when co-cultured in a Boyden chamber with neutrophils from WT or *MRP8-Bcl2* mice (Supplementary Fig. 2i). These data indicate that direct modification of target cells via cytokines from macrophages is not the major mechanism to activate PrCR, however there is a dynamic interplay that promotes macrophages to secrete CRT and neutrophils to accumulate it.

**Identification of CRT-binding targets**. We next sought to identify CRT-binding sites on neutrophils targeted for PrCR to better understand the basic mechanism of macrophage–target cell interaction during PrCR. CRT functions as a lectin-chaperone in the ER, binding to newly synthesized glycoproteins to assist their folding and glycosylation[32]. Abnormal cell-surface glycosylation has been reported on apoptotic cells and on some cancer cells[33,34]. Therefore, we hypothesized that macrophage-secreted CRT maintains its lectin-like characteristics and binds to target cells via cell surface glycoproteins, perhaps the same glycans CRT binds to in the ER[35,36]. Thus we performed a screening experiment by probing a carbohydrate microarray[37,38] that contains a large number of tumor glycan molecules, with recombinant CRT. We found that CRT selectively bound to the tumor-associated asialoglycans displaying the Tri-antennary and multivalent type II (Gal$\beta$1 → 4GlcNAc) chain epitopes (Tri/m-II) (Fig. 4a). Phytohaemagglutinin-L (PHA-L), a lectin that we previously demonstrated to specifically bind with the Tri/m-II dominant structure[37,38] (Supplementary Fig. 3a), mimics CRT-binding kinetics on neutrophils and macrophages, demonstrating that over time more binding sites for CRT can be detected on neutrophils as less are detected on macrophages (Fig. 4b). We found that while newly produced neutrophils displayed few CRT-binding sites as compared to mature neutrophils, treatment with neuraminidase, a glycoside hydrolase enzyme which removes sialic acids from the terminal positions of glycans and exposes the cryptic Tri/m-II, dramatically increased CRT binding in both groups (Fig. 4c, d). These data confirm that CRT can bind to the viable cells to promote PrCR via asialoglycoproteins. We then examined CRT binding to cell surface glycan molecules, by using WT and mutant CHO cell lines that specifically express Tri/m-II and its derivatives[39]. Stronger binding of recombinant CRT was observed on CHO lines defective in the CMP-sialic acid transporter (Lec2) or UDP-GlcNAc 2 epimerase (Lec3) that expressed Tri/m-II, as compared to parental line, or the line defective in UDP-Gal translocase that expressed Tri/m-II capped by sialic acids or missing mannose on the outermost layer (Supplementary Fig. 3b).

We next sought to determine whether the same mechanism functions for recognition and phagocytosis of target cells in PrCR during other biological processes. We have previously demonstrated that PrCR is critical for macrophage-mediated cancer immunosurveillance, so we examined the expression and roles of CRT-binding sites on cancer cells. We found that activated macrophages but not cancer cells were able to release a significant amount of CRT into the medium when cultured (Supplementary

Fig. 3c). In addition, we examined the presence of CRT-binding sites on a human AML cell line HL60 and a human colon cancer cell line SW620, and found that their endogenous CRT was either expressed at a low level or limited to the perinuclear regions while a significant portion of PHA-L staining were observed on the cell surface (Fig. 4e, f). Recombinant CRT bound to HL60 cells and colocalized with PHA-L on the cell surface, indicating that CRT is dependent on the presence of the cyptic Tri/m-II domain to bind (Supplementary Fig. 3d).

**Cell surface asialoglycans regulates CRT-mediated PrCR**. It has been shown that living aging and cancer cells can be directly phagocytosed by macrophages without cell death being induced[4,40] (Supplementary Fig. 4a, b). Clearance of living cancer cells through PrCR was not due to induction of cell death. During PrCR, cancer cells were engulfed and digested inside macrophages (Supplementary Fig. 4c–e). With the observation that CRT bridges target cells to macrophage for PrCR via asialoglycans on the cell surface, we wanted to next determine whether we can manipulate this asialoglycan epitope on any normal or neoplastic cell to promote PrCR. Indeed, we found that CRT binding to cancer cells was markedly enhanced after neuraminidase treatment and was specific to the asialoglycans (Fig. 5a–d); and that such treatment significantly promoted phagocytosis of a wide range of human cancer cells by either human or mouse macrophages (Fig. 5e, f and Supplementary Fig. 5a–c). To determine whether and which neuraminidase was responsible for promoting PrCR, we performed a screening experiment in which we suppressed the expression of the four different neuraminidase genes. We found that Neu4 played a major role in regulating cell surface asialoglycan epitopes for CRT binding to HL60 cells (Fig. 5g and Supplementary Fig. 5d–e), and consistently, suppression of Neu4 expression significantly decreased phagocytosis of these cells (Fig. 5h). Taken together these data demonstrate that the regulation of CRT-binding asialoglycans on target cells is critical for their susceptibility to PrCR, and can be produced with cell intrinsic neuraminidases.

**CRT-binding site in malignancies and hematopoiesis**. Since exposure of CRT-binding sites by neuraminidase treatment enhanced phagocytosis of cancer cells, we then examined its effects on inducing PrCR of xenotransplanted cancer cells in mice and reduction of tumor incidence in vivo. Importantly, treatment of cancer cells with neuraminidase showed no effects on the intrinsic programs regulating cell viability and proliferation (Supplementary Fig. 6a–f). When a human leukemia cell line (HL60) and a colon cancer cell line (DLD-1) were treated with neuraminidase and put into a long-term incubation with macrophages, significant reductions of cancer cell numbers were observed as compared to the control group (Supplementary Fig. 7a–b). When we subcutaneously injected HL60 and DLD-1 cells treated with active or heat-inactivated neuraminidase into NSG mice, a dramatic inhibition of tumor engraftment and growth was observed for the cells treated with active neuraminidase, suggesting the exposure of CRT-binding sites likely promoted PrCR in vivo (Fig. 6a, b and Supplementary Fig. 7c). Based on these findings, we reasoned that if the regulation of asialoglycans determines susceptibility to PrCR and therefore growth of tumor cells, then the expression of the genes that regulate the presence of asialoglycans could be correlated with disease outcomes or patient survival. To assess the correlation between the expression of CRT-binding sites and clinical outcomes, we used the PRECOG program[41] to analyze published gene datasets containing corresponding patient survival data. We included in this analysis genes encoding sialytransferases or neuraminidases, which respectively attach or remove sialic acid to

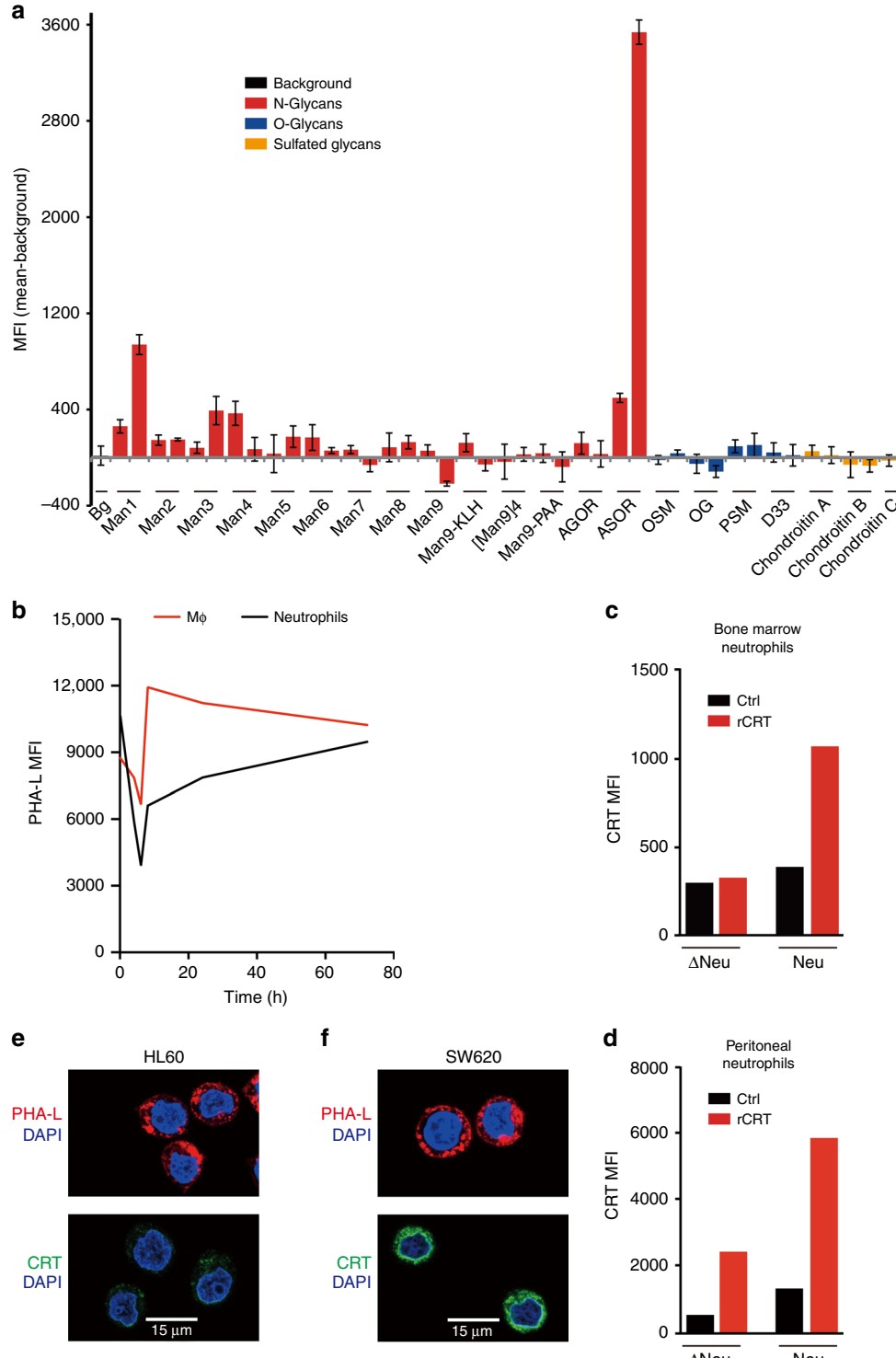

**Fig. 4** Identification of CRT-binding ligands on aged and malignant cells. **a** Screening for CRT-binding glycans with carbohydrate microarray. Purified CRT-IgG-Fc proteins were used to probe a carbohydrate microarray containing a number of different types of glycans, including N-, O- and sulfated-glycans. Anti-IgG-Fc antibody was used for detecting binding of CRT to glycans. Glyco-antigens were used at 0.05 and 0.25 μg μl$^{-1}$ (left and right bars for each glycan). $n = 3$. Error bars represent standard deviation. **b** PHA-L binding to peritoneal neutrophils and macrophages at 0, 4, 6, 8, 24, and 72 h after thioglycollate injection in *MRP8-Bcl2* mice. **c**, **d** Examination of cell surface CRT-binding sites on neutrophils. Bone marrow (**c**) or peritoneal neutrophils (**d**) were collected and treated with heat inactivated neuraminidase (Δneu) or neuraminidase (neu). Cells were incubated with PBS (control; black) or recombinant CRT proteins (red) and binding of CRT was then measured by flow cytometry with PE-conjugated anti-CRT antibody. rCRT binds to mature peritoneal neutrophils but not the immature bone marrow neutrophils. Treatment with neuraminidase led to the release of CRT-binding sites.

**e**, **f** Immunofluorescent staining of CRT in HL60 (**e**) and SW620 (**f**) cells. CRT localized to perinuclear regions, vesicles, and cell surface. CRT was either expressed at a low level (**e**) or limited to the perinuclear regions (**f**), while a significant portion of PHA-L staining were observed on the cell surface (**e** and **f**).In **a–d**, MFI, mean fluorescence intensity

glycoproteins, thus masking or unmasking Tri/m-II epitopes[42]. Higher expression of genes promoting the removal of sialic acids correlated with an improved survival[43,44] while higher expression of the genes enhancing sialic acid expression correlated with a worse outcome[45,46] (Fig. 6c–f).

We found previously that hematopoietic stem cells (HSCs) that migrated out of bone marrow and into circulation increased the expression of the "don't eat me" signal CD47, which blocked PrCR[1,13], suggesting stem cells also develop mechanisms against PrCR for their self-protection. Therefore, we expanded our

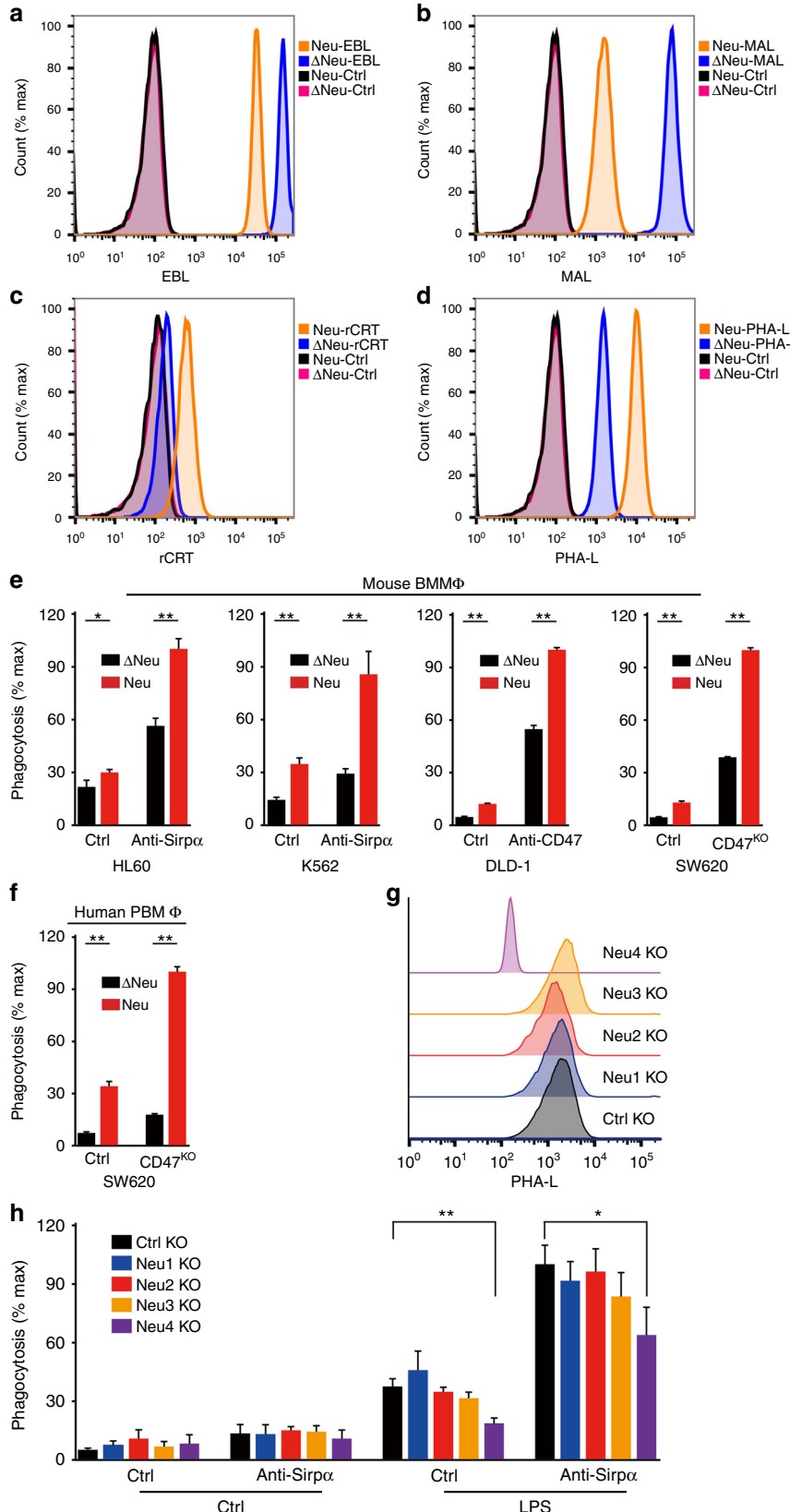

**Fig. 5** Cell surface asialoglycans regulates CRT-mediated PrCR. **a, b** Treatment with neuraminidase led to the removal of sialic acids from the cell surface of HL60 cells. HL60 cells were treated with heat inactivated neuraminidase (Δneu) or neuraminidase (neu). Cell surface sialic acids were examined by staining with EBL (a) and MAL (b) by flow cytometry analysis. EBL, Elderberry Bark Lectin; MAL, Maackia Amurensis Lectin II. **c, d** Examination of cell surface CRT and PHA-L binding sites on cancer cells. HL60 cells were treated with heat inactivated neuraminidase (Δneu) or neuraminidase (neu). Recombinant CRT (**c**) and PHA-L (**d**) binding after treatment were measured by flow cytometry. **e, f** Phagocytosis of cancer cells with neuraminidase treatment. In vitro Phagocytosis assays were performed with HL60, K562, DLD-1, and SW620 cells treated with heat inactivated neuraminidase (Δneu) or neuraminidase (neu) as target cells. Mouse bone marrow-derived (**e**) and human peripheral blood monocyte-derived (**f**) macrophages were used for the assay. Phagocytosis was normalized to the maximal response in the experiments. n = 3. *P < 0.05, **P < 0.01 (t-test) for phagocytosis between heat inactivated neuraminidase (Δneu)- and neuraminidase (neu)-treated groups. Error bars represent standard deviation. **g, h** Effects of suppressing the expression of endogenous neuraminidases in cancer cells. Neu1–Neu4 gene knockout were performed with CRISPR in HL60 cells. In vitro Phagocytosis assays were performed with HL60 cells as target cells and mouse bone marrow-derived macrophages (**h**). Macrophages were treated with PBS (ctrl) or lipopolysaccharide (LPS). Neu4 knockout led to the decrease of cell surface CRT-binding sites and inhibited cancer cell phagocytosis. Phagocytosis was normalized to the maximal response in the experiments. n = 3. *P < 0.05, **P < 0.01 (t-test) for phagocytosis between ctrl and Neu4 KO groups. Error bars represent standard deviation

findings to hematopoiesis and found further evidence for a role of these CRT-binding sites in PrCR in HSCs. In the current study, we used the Gene Expression Commons[47] platform containing gene expression datasets for different cell types in mouse and human to analyze expression of sialyltransferases and neuraminidases in HSCs and their differentiated progeny. HSCs showed significantly higher expression of sialyltransferases and lower expression of neuraminidases, consistent with their longer survival than their differentiated progeny (Supplementary Fig. 7d–k). We next examined CRT surface levels and availability of binding sites in acute myelogenous leukemia, as a pathological model of hematopoiesis. We fractionated patient samples into residual normal HSCs, multipotent progenitors (MPPs), leukemia stem cells (LSCs), and blasts cells. Importantly, staining of surface CRT and binding of PHA-L, significantly correlated in all cell populations, indicating the dependence of CRT cell surface binding on asialoglycoproteins in normal and neoplastic cells (Fig. 6g, Table 1 and Supplementary Fig. 8). Consistently, much stronger PHA-L staining was found on blast cells, which have the shortest half-life as compared to MPPs, LSCs, and HSCs (Fig. 6g).

## Discussion

Programmed cell removal is a common cellular process identified in many physiological and pathophysiological scenarios, and has recently been exploited in the treatment of malignancies by targeting CD47, which serves as a "don't eat me signal". CD47 blocking antibodies, now in clinical trials, have dramatic efficacy in the treatment of preclinical models of human diseases including patient-derived xenografts of cancers[1,17,48–51], fibrotic disorders[27], and atherosclerosis[28]. These antibodies function by restoring PrCR, with minor toxicity toward normal tissue cells.

The understanding of the underlying mechanisms of PrCR is still in its infancy. Whether and how the unwanted cells are programmed to be cleared is a critical question for the understanding of tissue homeostasis and treatment of multiple pathologic processes. Traditionally it has been believed that phagocytosis by macrophages is of dead cells, but our findings starting in 1994 show that even without PCD, phagocytosis by PrCR still occurs and homeostatically regulates cell numbers[3]. We have also demonstrated that many different human primary tumors resistant to PCD transplanted orthotopically in NSG mice are susceptible to activation of PrCR.

One of the key players in activation of PrCR is CRT, an eat me signal. We discovered here a CRT-mediated detection mechanism used by macrophages with stringent specificity that is responsible for the ability to distinguish wanted vs unwanted (damaged, dysfunctional, aged, or harmful) cells. We discerned that CRT is secreted by macrophages and binds to asialoglycoproteins on receptive target cells. Non-receptive cells have asialoglycoproteins that are "capped" with sialic acid that must be removed by neuraminidases before CRT is able to bind and initiate PrCR. Cancer cells express significantly higher level of both CRT-binding molecules and sialic acids than do normal cells[52,53], implying an evolving process during cancer development, in which cancer cells respond to internal signals, e.g., those that also induce cell death, to produce surface asialoglycoproteins, perhaps via secreted neuraminidases, or co-localization of neuramindase with post-golgi vesicles containing glycoproteins, and passively acquire "eat me" CRT signals. Successful cancer clones possess mechanisms to modify these signals or to express CD47 to evade PrCR. We have recently published that PDL1/PD1[54] and MHCIβ2/LILRB1[55] are also active as tumor ligands and macrophage receptors that provide "don't eat me" signals, pointing to the importance of this system. This new knowledge that CRT binds to asialoglycoproteins, which are often hidden by sialic acid, is the first step in dissecting the mechanism of PrCR initiation, which can be targeted in disease states such as cancer alone, or in combination with CD47 inhibition.

There are two fates for the neutrophil, PCD and/or PrCR; both have been demonstrated in vivo in mice in the natural circumstance. We showed in 1994 that wild-type neutrophils in response to sterile inflammation come into the peritoneal cavity from the bone marrow, peaking at 4 h[3]. At that time they are alive, and if incubated with macrophages, are not eaten. We now know they are CD47+ and CRT low. Between 12 and 24 h after thioglycollate administration, wild-type neutrophils are undergoing endonucleolytic degradation of their chromosomal DNA to nucleosome size fragments, but are not yet dead. If they are incubated with macrophages they are phagocytosed before they burst. The Bcl2+ neutrophils, which are resistant to PCD and are, therefore, not undergoing DNA degradation, are equally phagocytosed under inflammatory conditions when they are alive. All detectable neutrophils from either WT or Bcl2-expressing mice are gone by 24–48 h after thioglycollate-mediated inflammation. The rapid upregulation of CRT-binding sites on neutrophils within 24 h of maturation and recruitment to the peritoneum during induced sterile peritonitis implicates an increase in neuraminidase activity more than a decrease in sialyltransferase activity as a likely mechanism, insofar as the turnover of cell surface glycoproteins is slower[56,57]. An analogous phenomenon that reinforces this conclusion about factors contributing to PrCR and a shorter life-span of cells occurs in HSCs vs their differentiated progeny[58]. Here, the HSCs have a dramatic up- or down-

regulation, respectively, of sialytransferases and neuraminidases, leading to decreased CRT-binding sites and enhanced anti-PrCR

protection for these long-lasting stem cells, which is consistent with their remarkably low turnover[58,59].

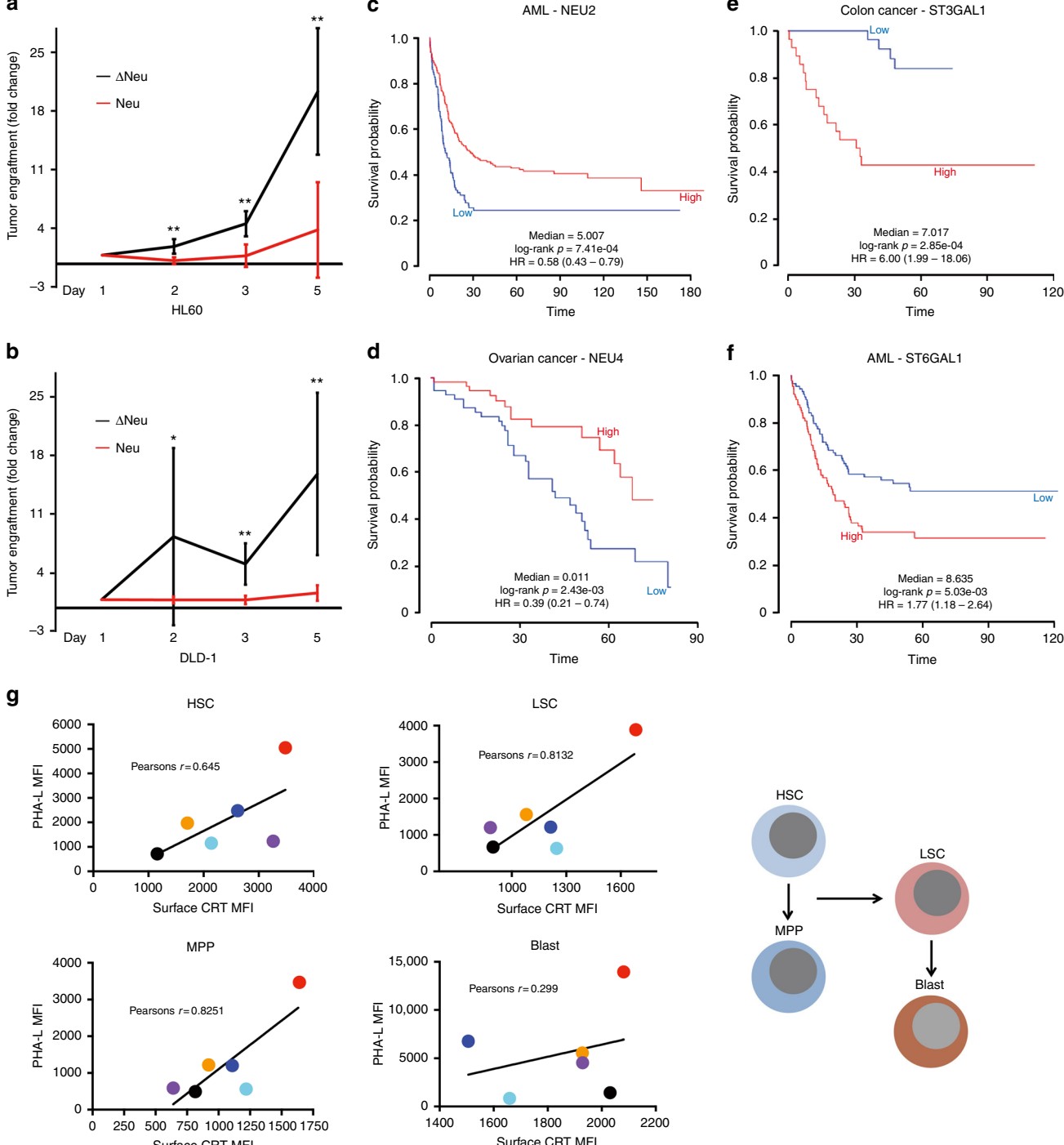

**Fig. 6** Function of CRT-binding site in multiple malignancies and hematopoiesis. **a**, **b** In vivo tumorigenicity of cancer cells treated with neuraminidase. HL60 and DLD-1 cells were treated with heat inactivated neuraminidase (Δneu) or neuraminidase (neu), and injected subcutaneously into NSG mice. Growth of transplanted tumors was monitored by bioluminescence imaging. Tumor growth was normalized to bioluminescence signals of the injection day as fold changes. $n = 5$. $*P < 0.05$, $**P < 0.01$ (t-test) for tumor growth between heat inactivated neuraminidase (Δneu)- and neuraminidase (neu)-treated groups. Error bars represent standard deviation. **c–f** Genes related to the regulation of CRT-binding sites as a diagnostic marker for overall survival of cancer patients. Higher expression of NEU2 and NEU4 which induce the removal of sialic acids correlated with an improved survival while higher expression of ST3GAL1 and ST6GAL1 which enhance sialic acid expression correlated with a worse outcome. **g** Correlation between PHA-L binding and CRT levels on the cell surface of hematopoietic stem cells (HSC), Multipotential progenitor (MPP), leukemia stem cells (LSC), and blasts cells from primary AML patient samples; analysis by flow cytometry. In **g**, MFI, mean fluorescence intensity

**Table 1 Clinical and molecular characteristics of primary human AML samples analyzed in Fig. 6g**

| Sample | SU050 | SU227 | SU369 | SU416 | SU444 | SU540 |
|---|---|---|---|---|---|---|
| Age | 50 | 30 | 86 | 67 | 60 | 68 |
| Gender | Male | Male | Male | Female | Male | Female |
| Primary/ secondary | Primary | Primary | Primary | Secondary | Primary | Primary |
| De novo/ relapsed | Relapsed | De novo | De novo | De novo | De novo | De novo |
| Cytogenetics | Normal | Normal | 48XY,+mar [3] | Normal | Normal | Normal |
| FLT3-ITD | Positive | Positive | Negative | Positive | Positive | Positive |
| %CD34 | <1% | 65% | <1% | 6% | 3% | 34% |
| WHO classification | AML with multilineage dysplasia without Antecede | AML with multilineage dysplasia without Antecede | AML-Not otherwise specified | AML with multilineage dysplasia following MDS | AML-Not otherwise specified | AML-Not otherwise specified |
| FAB | M1 | N/A | N/A | M5 | M4 | M4 |
| Somatic mutations | | CEBPA bi-allelic | NRAS Q61K | | DNMT3A R882H | NPM1c, DNMT3A R882H |

In the future, further studies are needed to address how, where, and by which mechanisms the asialoglycoproteins are produced in primary cancers, and whether distinct subgroups of macrophages are involved in "decoration" and engulfment of target cells, and finally how macrophages and other immune cells cooperate in immunosurveillance against asialgoglycan-mediated immune modulation[60,61].

In conclusion, we found during tissue turnover (differentiation, inflammation, etc.) and cancer development, cells regulate the accessibility of surface CRT-binding sites, which in turn controls their removal by macrophages through PrCR. Exposure of asialoglycoproteins leads to decoration of target cells with CRT, which is secreted by macrophages and eventually triggers PrCR (schematically summarized in Supplementary Fig. 9). Examination of CRT-binding sites on cancer cells demonstrated its prognostic power and a potential for creating new treatments by increasing the number of such sites to promote PrCR of a wide range of human cancer cells or other diseases. Understanding such mechanisms is critical for development of novel immunotherapies by fully exploiting PrCR. The elaboration of the PrCR signal [unopposed by CD47] in oligopotent hematopoietic progenitors in at least some myelodysplastic syndrome patients, in whom >95% of their HSC are from the single cell with a chromosomal anomaly such as monosomy 7, demonstrates that the PrCR signal can be elicited in a single differentiation step; the transition from MDS to AML is accompanied with upregulation of CD47[59]. Finally, a large number of human cellular degenerative diseases exist, and the mechanism of cell loss is generally speculated to be PCD; here, we provide another mechanism—PrCR—that also could be responsible.

There are other important implications from this study. First, formation of a mutant calreticulin is the critical lesion in up to half the patients with myeloproliferative neoplasms, MPN, and nearly all of the rest are due to a Jak2 activating mutation[62]. The first clue to this puzzle is that the mutant calreticulin is one that produces a frameshift in which the c-terminal domain loses $Ca^{2+}$ binding negative charged amino acids and replaces them with positive charged amino acids, and second, the frameshifted mutant CRT does not have a KDEL domain. The mutant calreticulin binds to the ectodomain of c-mpl, which signals via Jak2[62]. We propose that the binding of the CRT to c-mpl involves its lectin specificity, and that these c-mpl molecules are asialo-forms rather than sialylated proteins, and therefore the binding which leads to jak2 signalling could be wholly or in part due to the lectin domain of secreted CRT-binding asialo c-mpl in a way that triggers dimerization and signaling. Perhaps the new c-terminal domain also has c-mpl binding, but it may be that crosslinking asialoglycans on the protein is sufficient. This could imply that the MPN forms of CRT are dimers via the new c-terminal domain. It shall also be important to test whether the glycan portion of c-mpl is the CRT-binding site, and if the asialoglycan is also on other cell surface molecules, or only on c-mpl.

The second important implication of this study is the binding of PHA-L to the asialoglycans. PHA was discovered as an erythrocyte agglutinin which Nowell noticed also stimulated the non-agglutinated mononuclear cells in human peripheral blood, and in that situation stimulated the cells to enter cell cycle, which Nowell used to display the chromosomes of the individual for cytogenetic analyses[63]. PHA is a T-cell mitogen, and in the context of this paper, likely binds to asialoglycan residues on mitogenic T-cell surface molecules, now known to include the TCR, associated CD3 chains, and CD28. The erythrocyte specificity of PHA binding could also be via asialoglycans. We have shown elsewhere that cancer cells that are CD47+ can be phagocytosed in the presence of CD47 blocking antibodies, but only if the cells have displayed "eat me" signals such as CRT[64]. In mice, NHPs, and humans the first dose of therapeutic anti-CD47 antibodies results in a dose-dependent anemia that we interpret as due to saturation of RBC CD47[11,64] plus an "eat me" signal such as CRT, made by sinusoidal macrophages.

## Methods

**Mice.** BALB/c, RAG2$^{-/-}$ γc$^{-/-}$ BALB/c, FVB and NOD.Cg-Prkdc$^{scid}$ Il2rg$^{tm1Wjl}$/SzJ (NSG) mice were bred in a pathogen-free facility in the Institute for Stem Cell Biology and Regenerative Medicine at Stanford University and Animal Resources Center at City of Hope Comprehensive Cancer Center. Wild-type and Bcl-2 transgenic FVB mice were used for the isolation and analysis of peritoneal macrophages and neutrophils. For the transgenic mice, human Bcl-2 was expressed under the control of the human MRP8 regulatory regions[3]. All animal procedures were in accordance with the guidelines and approved by the Administrative Panel on Laboratory Animal Care at Stanford University and City of Hope Comprehensive Cancer Center.

**Cell culture.** Human cancer derived cell lines SW620, DLD1, HL60, K562, and murine macrophage/monocyte cell line J774 were obtained from ATCC. No additional cell authentication or mycoplasma examination was performed. Cells were routinely cultured in DMEM medium supplemented with 10% fetal bovine

serum (SW620, J774), IMDM medium supplemented with 20% fetal bovine serum (HL60), or RPMI-1640 medium supplemented with 10% fetal bovine serum (DLD1 and K562).

CHO and mutant lines identified as follows[39]: parent Gat-2, Lec2 (clone Gat-2.Lec2.4C, defective in the CMP-sialic acid transporter), Lec3 (clone Gat-2.Lec3.6F, defective in UDP-GlcNAc 2 epimerase) or Lec8 (clone Gat-2.Lec8.1C, defective in UDP-Gal translocase). All cells were cultured in alpha-MEM (Gibco 11900−073) supplemented with 10% Fetal Calf Serum. Cells were maintained in suspension or monolayer.

**Plasmids**. A pCDH-CMV-MCS-EF1 lenti viral vectors expressing a luciferase-eGFP fusion protein or a luciferase-RFP fusion protein were transfected to 293T cells to generate lenti viruses, which were used to transduce tumor cells (SW620, DLD1, HL60, K561, with the eGFP lenti viruses) or J774 macrophage cells (with the RFP lenti viruses). Cells were then sorted by flow cytometry with BD FACSAria II cell sorters for GFP+ or RFP+ cells[17]. A pCEP4 episomal mammalian expression vector (Thermo Fisher) was used for the expression of recombinant CRT-IgG-Fc proteins.

**Generation of macrophages**. To generate mouse bone marrow-derived macrophages, bone marrow cells were isolated from NSG or RAG2$^{-/-}$, $\gamma$c$^{-/-}$ mice that were 6–10 weeks old. The cells were treated with ACK lysis buffer for 5 min and then cultured in IMDM medium supplemented with 10% FBS and 10 ng ml$^{-1}$ of MCSF for 6–8 days for them to differentiate to macrophages.

To generate human peripheral blood-derived macrophages, monocytes were enriched from human peripheral blood (purchased from Stanford Blood Center) and cultured in IMDM supplanted with 10% human serum for 7–10 days for them to differentiate to macrophages.

**Isolation of neutrophils and macrophages**. Peritoneal lavage was conducted at indicated times after injection of 2 ml of 3% thioglycollate medium (Difco) i.p. Neutrophils were collected by from the lavage at indicated times, washed with RPMI 1640 and macrophages were allowed to adhere to 24-well plates in RPMI 1640 medium for 2 h and non-adherent cells were collected. Forty-eight hours after injections peritoneal cells were collected and were used for downstream applications. Thioglycollate elicited macrophages were cultured at $0.5 \times 10^6$ cells per well in RPMI-1640 supplemented with 1% serum in a 24-well plate. Macrophages were co-cultured with neutrophils using a 0.4 µm pore transwell (corning life sciences) with neutrophils cultured in the upper chamber. Cells or supernatant were collected 16–24 h after co-culturing.

Bone marrow from MRP8−Bcl2 mice were collected and crushed and layered over 1077 and 1119 histopaque then centrifuged. Cell layers containing monocytes/macrophages or neutrophils were recovered and washed into PBS then fixed for 20 min with 4% PFA at room temperature. Cells were then blocked for 1 h with PBS containing 1% FCS at room temp followed with staining for calreticulin overnight at 4 °C. Cells were washed three times and AF488 secondary was added and incubated at RT for 1 h then washed three times with wash buffer containing DAPI. Cells were then imaged on the confocal microscope.

**Flow cytometry analysis and sorting**. Anti-CD47 (anti-mouse CD47, 127510, Biolegend; anti-human CD47, 17-0479-42, eBiosciences), anti-calreticulin (ADI-SPA-601PE-F, Enzo Life Sciences; ab22683, Abcam; polyclonal anti-CRT JM-3077-100, MBL International), anti-F4/80 antibodies (123112, 123128 Biolegend; 565411, BD Biosciences), anti-Gr-1 antibody (108422, Biolegend), anti-CD11b (anti-mouse CD11b, 563402, 563553, 563168, BD Biosciences; anti-mouse CD11b, 56-0112-80, eBiosciences; anti-human CD11b, 301412, 101257, Biolegend; anti-human CD11b, 555389, BD Biosciences), anti-CD2 (555328, BD Biosciences), anti-CD19 (555414, BD Biosciences), anti-CD20 (555624, BD Biosciences), anti-CD90 (562556, 555595, BD Biosciences), anti-CD38 (335790, BD Biosciences), anti-CD45RA(560362, 304118, BD Biosciences) were used for FACS analyses. Antibodies were Phycoerythrin (PE)-, PE cy-7-, PE-Cy5, V450, FITC, APC-, or Brilliant Violet 421 (BV421)-conjugated, or fluorophore-conjugated secondary antibodies were used. Sytox blue or 7-AAD was used to exclude dead cells. Cells were identified by surface markers, such as CD11b$^+$F4/80$^-$GR1$^{++}$ (Neutrophils) and CD11b$^+$F4/80$^+$ (macrophages).

Human AML samples were used and populations were identified such as HSC (Lin$^-$CD34$^+$CD38$^-$CD90$^+$CD45RA$^-$), MPP (Lin$^-$CD34$^+$CD38$^-$CD90$^-$CD45RA$^-$), LSC (Lin$^-$CD34$^+$CD38$^-$CD90$^-$), and Blasts (Lin$^-$CD34$^-$)[65]. Cells were blocked with either IV.3 or 2.4G2 for 20 min on ice then stained with the appropriate markers for 30 min on ice, washed and acquired or sorted on flow cytometer. FACSDIVA software was used while acquiring and sorting cells and post analysis was conducted with FlowJo 10.2 software. Human AML samples were obtained from patients at the Stanford Medical Center with informed consent, according to institutional review board (IRB)-approved protocols (Stanford IRB, 18329 and 6453).

For CRT-binding experiments, cells were incubated with recombinant calreticulin human Fc fusion protein at 40 µg ml$^{-1}$ or biotin-PHA-L 6 µg ml$^{-1}$ (B-1115, Vector laboratory) for 1 h incubated on ice. Cells were washed three times then incubated with anti-human Fc-AlexaFluor647 (309-605-008, Jackson

ImmunoResearch) or Streptavidin-AlexaFluor488 (s11223, Invitrogen) for 30 min on ice. Cells were washed twice and co-stained with DAPI (Sigma) then relative fluorescence was analyzed on the flow cytometer (BD Fortessa).

**RNAseq analysis**. Wild type and transgenic MRP8-Bcl2 mice were injected with 2 mL thioglycolate to induce inflammation. Four or eight hours after injection, cells extracted from the peritoneal cavity and bone marrow of each mouse were FACS sorted directly into Trizol prior to total RNA extraction with chloroform and purification with the Qiagen RNeasy MinElute Cleanup Kit. From the total RNA, cDNA synthesis and amplification was carried out using NuGen's Ovation RNA-Seq System V2, sheared using Covaris, and size-selected using Agincourt AMPure XP beads.

For library preparation, we used the NEBNext Ultra RNA Library Prep Kit for Illumina, using Illumina TruSeq Adapters for barcoding. Depending on the concentration of adaptor-ligated cDNA, as determined by Qubit quantitation, 6–15 PCR cycles were carried out to amplify the libraries. Agincourt AMPure XP beads were used to purify the adapter-ligated cDNA post-amplification. The samples were then multiplexed and data were generated on an Illumina HiSeq 4000 using paired-end $2 \times 75$ bp read lengths. The data were analyzed using AltAnalyze[66], with analysis parameters set to default values and pathway analysis was conducted used PANTHER[67].

**Quantitative PCR**. Total RNA was extracted as described above and cDNA was generated using superscript III (Invitrogen) and quantitative PCR was conducted with SYBR Green (Applied Biosystems) and acquired on the ABI 7500.
The following primer sequences were used:
GAPDH Fwd: ACCACAGTCCATGCCATCA
GAPDH Rev: CACCACCCTGTTGCTGTAGCC
CRT Fwd: GCAGACCCTGCCATCTATTTC
CRT Rev: TCGGACTTATGTTTGGATTCGAC

**Phagocytosis assay**. In vitro phagocytosis of neutrophils and tumor cells by macrophages were quantified with FACS-based assays. Differentiated mouse bone-marrow-derived macrophages and human peripheral blood-derived macrophages were detached from culture dishes with TrypLE and scrapers, and divided into FACS tubes or low-attachment 96-well plates, with $1–5 \times 10^4$ cells per well per tube. Tumor cells were pre-labeled with eGFP lenti viruses and neutrophils were stained with Cell-trace Calcein dye prior to experiment. Target cells (tumor cells or neutrophils) were added and mixed with macrophages in IMDM medium, and incubated at 37 °C for 2 h with indicated conditions (antibody/enzyme treatment). For blockade of CD47-Sirpa interaction, anti-CD47 (B6H12, BD Biosciences) or anti-Sirpa (BioLegend) antibodies were used. Anti-mouse F4/80 antibody was used to stain macrophages. Phagocytic index was quantified by FACS analyses and calculated with the number of macrophages that phagocytosed target cells (F4/80+GFP+)/divided by the number of total macrophages (F4/80+). In each experiment, phagocytic indexes were normalized to the maximal indexes. Alternatively, cells were loaded with LysoTracker Red DND-99 probes (Thermo Fisher Scientific) in growth medium at 37 °C for 1 h. Cells were then washed once with growth medium and twice with IMDM medium, and incubated with macrophages for phagocytosis. Excitation and Emission for LysoTracker probes are 577 and 590 nm, respectively. LysoTracker probes are excited in a PH-dependent manner and are highly selective for acidic organelles. Cancer cells loaded with LysoTracker probes become positive in PE channel when they are engulfed and delivered to acidic organelles in macrophages for digestion. After incubation with target cells, macrophages that are F4/80+PE+ are the ones that engulfed target cells.

To evaluate the long-term effect of phagocytosis, macrophages and target cells were mixed and co-cultured in 24-well plates for 24 h with indicated conditions (antibody/enzyme treatment). Cells were collected by TrypLE and anti-mouse F4/80 antibody was used to stain for macrophages. After incubation, cells were washed with FACS buffer and resuspended with FACS buffer containing sytox blue. Non-colored standard cells (293T cells were used as standard cells) were added prior to FACS analyses. Remaining target cells that surviving from phagocytosis by macrophages were normalized to standard cells (numbers of standard cells were fixed and equal in each sample) to evaluate the effects of phagocytosis at different conditions.

**Cell viability measurement**. To measure cell viability, neutrophils, macrophages, or cancer cells were washed twice with PBS, and stained with AnnexinV in AnnexinV-binding buffer for 15 min at room temperature in the dark. AnnexinV-binding buffer containing cell viability dyes (DAPI or sytox blue) were added and cells were analyzed by flow cytometry analyses.

**Immunofluorescence imaging**. Cells were collected, fixed, and permeabilized following the manufactures recommendations (BD Cytofix/CytoPerm). Cells were then blocked with 5% BSA in PBS and stained for calreticulin (MBL) at 1:500 overnight at 4 °C. Cells were then washed three times with PBS containing 1% tween 20 by resuspending cells and pelleting the cells by centrifugation at $200 \times g$ for 5–10 min. Secondary Alexa fluor 488 (Invitrogen) antibodies were

then used at 1:1000 and cells stained for 1 h at room temperature. Cells were washed three times and DAPI was used to stain nuclei at 300 nM (Invitrogen). Confocal images were acquired using the Zeiss LSM710 and ImageJ was used for overlaying images.

**Click-iT assay.** Thioglycolate-induced macrophages were isolated as described above. Macrophages were plated at $0.5 \times 10^6$ cells per well in a 24-well plate and were labeled with Click-iT AHA (Thermo Fisher) overnight in DMEM L-methionine free (Invitrogen) + 1% serum. Macrophages were washed then co-cultured with neutrophils isolated as described above after induction of peritonitis. Neutrophils were co-cultured with macrophages using a 0.4 μm pore transwell for 16 h. Neutrophils were washed three times with PBS. Cells were then lysed in lysis buffer (20 mM Tris-HCl, pH 7.4, 150 mM NaCl, 2 mM EDTA, supplemented with 1% Triton X-100, protease inhibitor cocktail and phosphatase inhibitor cocktail) with brief sonication. Cell lysate was incubated for 1 h with GammaBind Plus Sepharose for preclearance and overnight with anti-CRT antibodies at 4 °C. GammaBind sepharose was added to cell lysate and incubated for 1 h at 4 °C. Beads were washed three times with lysis buffer. Immunoprecipitated proteins were eluted with PBS supplemented with 1% SDS, incubated with biotin alkyne (Thermo Fisher) and labeled with the Click-iT protein reaction buffer kit (Thermo Fisher). The samples were then subjected to SDS-PAGE and western blot. HRP-conjugated streptavidin was used to detect biotin-labeled proteins. For loading controls, lysate from neutrophils or macrophages incubated with the methionine analog for immunoprecipitation was reduced and loaded onto a SDS-PAGE gel and then stained with Sypero Ruby protein gel stain to visually verify total protein content of samples.

**ELISA.** Supernatants were collected at the indicated times and first centrifuges at 200×g for 5 min at 4 °C then centrifuged at 1000×g for 10 min at 4 °C then stored immediate at −80 °C. Calreticulin was measured using an ELISA kit from LSBio and following the manufactures instructions. Supernatants were also assessed for soluble factors by Luminex assays by the Stanford Human Immune Monitoring Center.

**CRISPR-based gene knockout.** Suppression of gene expression was performed using CRISPR. Pairs of primers containing sgRNA sequences targeting human Neu1–Neu4 genes were designed and cloned into the all-in-one LentiCRISPR V2 plasmid[68]. The lentiviral vectors were transfected together with packing plasmids to 293T cells. Virus was collected 48 h after transfection and added to target cells. Target cells were selected for infection with puromycin after 48 h incubation with virus.

The following sgRNA sequences were used:
Control (LacZ): UUGGGAAGGGCGAUCGGUGC[69]
Neu1: GACCCCACUUCCGUAGCGCC[68]
Neu2: CCCCGUCUGCGCGUCUCACA[68]
Neu3: UUGCAGAGAAGCGUUCUACG[68]
Neu4: UCAUGGACCGGUGCUCCGC[68]

**Carbohydrate microarrays.** Carbohydrate antigens of various complexities were dissolved in PBS (glycoprotein conjugates) or saline (polysaccharides), and spotted onto SuperEpoxy 2 Protein slides (ArrayIt Corporation, Sunnyvale, CA). Immediately before use, the printed microarray slides were washed in 1X PBS at RT for 5 min, and blocked with 1% BSA-PBS at RT for 30 min. They were then incubated with recombinant CRT-IgG-Fc (0.05 μg μl$^{-1}$) at RT for 1 h followed by washing and then incubated with Cy5-tagged goat anti-mouse IgG-Fc secondary antibodies at RT for 30 min. The stained slides were rinsed five times and spin-dried at room temperature before scanning for fluorescent signals. The ScanArray5000A Microarray Scanner (PerkinElmer Life Science) was used to scan the stained microarrays. Fluorescent intensity values for each array spot and its background were calculated using ScanArray Express software (PerkinElmer Life Science).

**Generation of CRT recombinant proteins.** The FreeStyle 293F cells (Thermo Fisher Scientific) were used for expressing CRT recombinant proteins according to manufacturer's instructions. Briefly, a vector expressing hCRT-IgG-Fc was transfected to 293F cells with 293 fectin (1 ml per 500 μg of plasmid DNA). Cells were cultured in FreeStyle 293 Expression Medium (Thermo Fisher Scientific) in a 37 °C incubator with 8% CO$_2$ on an orbital shaker platform rotating at 130 r.p.m. A total of 5–7 days after transfection, the supernatant was collected to flow through the HiTrap Protein A HP column. Purified proteins were eluted from the column with 0.1 M Glycine (pH 2.7) and neutralized with 1 M Tris-buffer (pH 8.0). The eluted proteins were then buffer changed to PBS (pH 7.4) by repeated centrifugation with ultracentrifugal filter concentrators (Millipore).

**Neuraminidase treatment.** Cells (neutrophils or tumor cells) were washed with PBS and the incubation buffer (DMEM medium supplemented with 2 mM CaCl$_2$ and 0.25% BSA). A total of $1–5 \times 10^6$ cells were then treated with 0.5 U ml$^{-1}$ Vibrio Cholerae in the incubation buffer at 37 °C for 2 h. After the incubation, cells were

washed twice with IMDM medium and subjected to phagocytosis assay and FACS analyses.

**Tumor engraftment and treatment.** HL60 and DLD1 cells treated with neuraminidase or heat-inactivated neuraminidase were washed twice with DMEM medium and suspended in DMEM medium with 25% matrix matrigel. Cells were then injected subcutaneously on the back of NSG mice that were 6–10 weeks old. Tumor cell survival and growth in vivo were monitored by bioluminescent imaging, as described before[17]. Briefly, D-Luciferin (firefly) potassium salt was dissolved in PBS to a final concentration of 16.6 mg ml$^{-1}$ and injected intraperitoneally to mice with a dose of 0.139 g luciferin kg$^{-1}$ body weight. Bioluminescent imaging was performed daily or once every 2 days. Mice were imaged and tumor signals were analyzed with Living Image 4.0 software.

**Statistical analysis and graphing software.** All data were collected and statistically analyzed and graphed with either Microsoft excel or Prism graphpad software.

**Data availability.** The RNA-seq data reported in this paper have been deposited in the Gene Expression Omnibus database under the accession code GSE95631. The authors declare that all the other data supporting the findings of this study are available within the article and its Supplementary Information files and from the corresponding author upon reasonable request.

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

## Acknowledgements

We thank R. Majeti, A. McCarty, S. Karten, T. Storm, J. Seita, K. Loh, N. Fernhoff, R. Lu, A. Newman, R. Sinha, T. Naik, and T. Raveh for scientific discussions, technical assistance, and reagents; J. Seita for provision of the gene expression commons; P. Lovelace and J. Ho for assistance with FACS; and L.X. Wang and the Kabat Collection of Carbohydrate Antigens at SRI International for a number of carbohydrate antigens that were applied in this study. Flow cytometry analysis was performed at the FACS facility in the Institute for Stem Cell Biology and Regenerative Medicine at Stanford University, and the Analytical Cytometry Core at City of Hope supported by the National Cancer Institute of the National Institutes of Health under award number P30CA033572. The primary AML samples were obtained from the Stanford University Division of Hematology Tissue Bank. This work was supported by The Virginia and D.K. Ludwig Fund for Cancer Research, the National Cancer Institute of the National Institutes of Health under award numbers R01CA086017 (to I.L.W.), P01CA139490 (to I.L.W.), K99/R00 Pathway to Independence Award K99CA201075/R00CA201075 (to M.F.), T32DK098132 (to K.D.M.), and R56AI118464 (to D.W.), the Damon Runyon-Dale F. Frey Award for Breakthrough Scientists DFS-22-16 (to M.F.), SRI International Proposal No. 15-

141BDH (to D.W.), and Koshland Integrated Natural Science Center Summer Scholars Program at Haverford College (to K.S.). Funds from an anonymous donor helped accelerate these studies. This paper illustrates the power of glycosciences in solving major problems in biomedical research[70].

## Author contributions

M.F., K.D.M., Z.F. and R.W.-T. are co-first authors and A.L., K.S. and K.S.K. are co-second authors. K.D.M. may list himself first in his CV. M.F., K.D.M., and I.L.W. designed the research; M.F., K.D.M., F.Z. and R.W.-T. performed and analyzed the experiments, with suggestions from I.L.W.; A.L., K.S., K.S.K, M.M., P.A.B, H.M.J., T.S.C, J.C., A.M.B., P.G. and L.C. performed experiments with assistance of M.F. and K.D.M.; J.L. generated CRT recombinant proteins; D.W. conducted the carbohydrate microarray screens; M.F. and K.D.M. prepared the figures and wrote the manuscript; I.L.W. edited the manuscript and supervised the laboratory.

## Additional information

**Competing interests:** M.F. declares patent applications pertaining to stimulating TLR/BTK signaling to promote CRT in macrophages assigned to the Stanford University and equity and/or consulting with Forty Seven, Inc. I.L.W. is a cofounder and member of the board of directors for Forty Seven, Inc., a company that is developing therapies involving the CD47-SIRPα axis. P.G. and J.L. are currently employees of Forty Seven, Inc. The remaining authors declare no competing interests.

