## [Peer Review File · Nature Communications]

Reviewers' comments:

Reviewer #1 (Remarks to the Author):

The MS programed cell removal by calreticulin in tissue homeostasis and cancer by Feng et al, supports a potentially important role for extracellular calreticulin in identifying and promoting phagocytosis of living cells (in inflammation and cancer) and shows asialoglycans on the target cells to represent the relevant binding sites for the calreticulin. This is an important novel observation and is in general, adequately supported by the data presented in the MS. On the other hand, the paper is huge, hard to read and many of the figures hard to interpret due to lack of clear descriptions. The data hop from in vivo, to ex vivo to in vitro often without clear indication of which. Since the in vivo studies involve not only changes in the cells that have accumulated, but also constantly changing numbers of new cells (as well as loss of existing ones) some of the time-course data presented may be very difficult to interpret. Also, many figures express the data as "% reactive cells" or "% of maximum phagocytosis" or "fold change" in each case not indicating change from, or percentage of, what? Numbers of experiments, statistics etc are often lacking (for example in Figs 2a,b and c) One further concern is that the neutrophils (an important part of the whole story) appear to have been identified by GR1 – which also identifies a population of monocytes.

Reviewer #2 (Remarks to the Author):

In this manuscript the authors Feng M et al. describe a novel role for calreticulin in cell removal by macrophages. The authors show that calreticulin is released by macrophages which is involved in the removal of living cells. Moreover, the role of asialoglycans and neuraminidase-4 in phagocytosis have been proposed. This study proposes a general mechanism for the removal of unwanted cells in physiological and pathophysiological processes. In general it is an interesting concept that viable cells can be removed by antigen presenting cells. However, I think that this study is missing some important controls and more thorough discrimination between viable and dead cells should be performed. The type of cell death should be better identified. The data on the fact that viable cells can be phagocytosed are not convincing. Therefore additional experiments have to be address to confirm the main conclusions of the manuscript. The manuscript can be considered for publication in the journal after a major revision.

Major comments:

- the authors used to define apoptosis Annexin-V and DAPI staining (Fig. 1b). Of note, that this staining is not specific for apoptosis because other cell death modalities for example like necrotic cells can also expose PS on their surface at the same moment can be stained with Annexin-V while remaining PI- and with preserved plasma membrane (PMID: 28650960, PMID: 28388412). So it is advised to use other methods to characterise the type of cell death. For example caspase-8/PARP WBs, p-MLKL.

- How sure can be the authors that necroptosis is not involved?

- Also it is not clear what the authors understand under viability of cells, see Fig 1B (AnV+DAPI+?). Clarifications should be done.

- Fig. 2D: the authors analyse phagocytosis of cells. The cell death analysis from the same experiments should be presented. How many AnV+/PI+, AnV+PI-, AnV-PI+ and AnV-PI- cells were given for the uptake with macrophages.

- Fig 3: the authors demonstrate a comparative analysis of CRT expression in macrophages and neutrophils. Since the PI+ cells can be also positive for CRT, it is advised to exclude such cells from the analysis. In other words what is the % of viable cells in these experiments (Fig. 3A, 3D,

3E). These data should be included in the manuscript.

- What was the % of dead/viable cells in the phagocytosis experiments presented in the Fig. 5E, 5F, 5H, 5J). These data should be presented.

- Fig. 6: the authors should compare the cell death rate in vivo in the tumour. Since it is conceivable the cells are just dying more rapidly in 'Neu' condition.

- it is often very difficult to analyse phagocytosis of dead cells by flow cytometry because discrimination between attached cells and engulfed cells is quite problematic. For this pH sensitive probes for phagocytosis should be used. Please prove that you indeed measure phagocytosis and not cell attachment.

- isotope control ABs should be included in all experiments (e.g. Figs 2A,2B, 2C; Fig3).

Reviewer #3 (Remarks to the Author):

In this report the authors identify the endoplasmic reticulum protein calreticulin as a programmed cell removal (PrCR) signal. It was picked up as a candidate PrCR from a screen of activated macrophages from >300 genes that showed increased expression upon activation by toll like receptor ligands. Evidence suggests that CRT is transported to the cell surface of macrophages, and can then 'label' activated neutrophils by binding to asialo-glycoproteins, which are generated by the human neuraminidase, Neu4. The CRT then acts as an 'eat me' PrCR signal for macrophages.

While the proposal is interesting, and the data presented suggest how CRT can work as a factor in PrCR, it is not compelling. The rationale for production of CRT by macrophages and decoration of cells displaying 'eat me' signatures, in this case asialoglycoproteins, supports a consistent story, but each piece of evidence by itself is supportive in some cases and weak in others.

1. Of the many activated neutrophil genes that can be inferred to be relevant to PrCR (Fig. 1), it is not clear how CRT was selected for further study.
2. The correlation of CRT expression on macrophages and neutrophils over time following thioglycolate treatment is done without statistics (Fig. 2a-2c).
3. Data suggesting that CRT recognizes asialoglycoproteins relies heavily on binding to a glycan array result showing that it selectively binds to asialo-orosomucoid (ASOR), which is a serum glycoprotein, alpha 1 acid glycoprotein. CRT has been run on the much larger glycan arrays of the Consortium for Functional Glycomics by several groups (functionalglycomics.org), and results were inconclusive (no binding) or no binding to galactose terminated glycans, including tri-antennary N-linked glycans terminated with galactose.
4. Use of exogenous neuraminidases to show changes in lectin binding and phagocytosis in Fig. 5 are correlative, but exogenous bacterial neuraminidases dramatically desialylate surface glycoproteins and create artifacts that may or may not be connected to the biology being studied. There is no direct connection to Neu4 or CRT biology.
5. Use of oseltamivir as a neuraminidase inhibitor in Fig. 5g&h is puzzling, since this is an influenza virus inhibitor that has been shown not to inhibit human Neu1, 2, 3, or 4.
6. Use of PHA to detect binding sites for CRT in neuraminidase treated HL60 cells (Fig. 6a&b) is overstating the conclusion that CRT has the same specificity as PHA. This has not been shown.
7. While the correlation of survival with decreased expression of sialyltransferases or increased expression of Neu4 adds biological significance, the association of high sialic acid and poor prognosis has been around for decades, and there is no clear tie to the CRT/PrCR mechanism for this correlation.

Response to Reviewers' comments:

Reviewer #1 (Remarks to the Author):

The MS programed cell removal by calreticulin in tissue homeostasis and cancer by Feng et al, supports a potentially important role for extracellular calreticulin in identifying and promoting phagocytosis of living cells (in inflammation and cancer) and shows asialoglycans on the target cells to represent the relevant binding sites for the calreticulin. This is an important novel observation and is in general, adequately supported by the data presented in the MS. On the other hand, the paper is huge, hard to read and many of the figures hard to interpret due to lack of clear descriptions. The data hop from *in vivo*, to *ex vivo* to *in vitro* often without clear indication of which. Since the *in vivo* studies involve not only changes in the cells that have accumulated, but also constantly changing numbers of new cells (as well as loss of existing ones) some of the time-course data presented may be very difficult to interpret. Also, many figures express the data as “% reactive cells” or “% of maximum phagocytosis” or “fold change” in each case not indicating change from, or percentage of, what? Numbers of experiments, statistics etc are often lacking (for example in Figs 2a, b and c) One further concern is that the neutrophils (an important part of the whole story) appear to have been identified by GR1 – which also identifies a population of monocytes.

We thank the reviewer for the very positive comments. We have revised the manuscript as the reviewer suggested, and we have added necessary information to the figure legends. *In vitro* and/or *in vivo* experiments are indicated in the figure legends. “% reactive cells” or “% of maximum phagocytosis” and “fold change” have been clarified in figure legends. We have now included figures which incorporates multiple mice and statistical analysis for each time point for Fig2a, b, and c. Isotype controls were labeled in the figures. We consistently observed the same trend for CRT, CD47 and rCRT staining for neutrophils and macrophages after thioglycollate injection, while we did observe some slight differences between different individual experiments including our initial graphs in MFI values, mostly due to variations between individual mice and experimental settings. We include in the manuscript a representative experiment from at least 3 mice for each time point.

In the experiments with neutrophils and macrophages, the cells were collected from peritoneal lavage and majority of the cells were neutrophils and macrophages in this case (Lagasse and Weissman, 1994, Hoover and Nacy, 1983, Watt et al, 1979, Luo and Dorf, 2001). We then gated neutrophils based on GR1^{high}F4/80⁻, further analysis via F4/80 expression indicates that this double positive population is CD11b^{high}. The sorted populations using the gating strategy resulted in cells with morphology of neutrophils or macrophages via microscopy.

Reviewer #2 (Remarks to the Author):

In this manuscript the authors Feng M et al. describe a novel role for calreticulin in cell removal by macrophages. The authors show that calreticulin is released by macrophages which is involved in the removal of living cells. Moreover, the role of asialoglycans and neuraminidase-4 in phagocytosis have been proposed. This study proposes a general mechanism for the removal of unwanted cells in physiological and pathophysiological processes. In general it is an interesting concept that viable cells can be removed by antigen presenting cells. However, I think that this study is missing some important controls and more thorough discrimination between viable and dead cells should be performed. The type of cell death should be better identified. The data on the fact that viable cells can be phagocytosed are not convincing. Therefore additional experiments have to be address to confirm the main conclusions of the manuscript. The manuscript can be considered for publication in the journal after a major revision.

Major comments:

- the authors used to define apoptosis Annexin-V and DAPI staining (Fig. 1b). Of note, that this staining is not specific for apoptosis because other cell death modalities for example like necrotic cells can also expose PS on their surface at the same moment can be stained with Annexin-V while remaining PI- and with preserved plasma membrane (PMID: 28650960, PMID: 28388412). So it is advised to use other methods to characterise the type of cell death. For example caspase-8/PARP WBs, p-MLKL.

We thank the reviewer for this suggestion. This study was to explore the molecular mechanisms of programmed cell removal (PrCR), which plays essential roles in many basic biological processes. While PrCR is a critical step of dealing with cells undergoing programmed cell death (PCD), in the manuscript, we are focusing on the mechanisms of PrCR in clearance of viable cells. So we used the best studied model of sterile inflammation to test if PrCR was responsible for removing dying cells, and to test if neutrophils that can't undergo PCD due to enforced expression of Bcl2 were still removed by PrCR; that is, what are the roles of PCD vs PrCR in neutrophile lifespan? Thioglycolate-induced acute inflammation is one of the most established models for PrCR – we have previously showed that, using the MRP8-Bcl2 transgenic mouse model, when PCD was blocked by expressing the anti-apoptotic protein Bcl2 [and we showed that the Bcl2+ cells don't die during the experiment], PrCR still occurred.

We have showed previously when anti-phagocytic “don't eat me” signals were blockade, living cancer cells were phagocytosed directly without undergoing apoptosis first (Jaiswal et al. 2009; Majeti et al. 2009; Chan et al. 2009; Chao et al. 2010; Chao et al. 2011; Willingham et al. 2012; Edris et al. 2012; Weiskopf et al. 2013; Feng et al. 2015; Weiskopf et al. 2016; Barkal et al. 2018; etc.). We now added new figures (Supplementary Fig. 2d, Supplementary Fig. 4a&4b, Supplementary Fig. 6a-6e) by staining cells with AnnexinV and Sytox Blue/DAPI, showing that the majority of the

target cells (>95%) stayed alive before and after phagocytosis assay unless they were engulfed by the macrophages.

Again, we thank the reviewer for the suggestion and agree that it is important to understand the relevance between different types of cell death and PrCR. However, we mainly focused on PrCR of living cells in this manuscript and we didn't induce cell death in our models. We have revised our manuscript to make this point clearer.

- How sure can be the authors that necroptosis is not involved?

In this manuscript, we mainly focused on PrCR of living cells and we didn't induce cell death in our models. It is possible both apoptosis and necroptosis as well as other cell death occurred, but majority of the cells we used in our assays were viable (>95%). We are focusing on understanding the mechanisms of PrCR of these viable cells. We have revised our manuscript to make this point clearer, and we have changed "apoptosis" in the manuscript to "cell death" to be more accurate.

- Also it is not clear what the authors understand under viability of cells, see Fig 1B (AnV+DAPI+?). Clarifications should be done.

We used AnnexinV and DAPI to gate cells undergoing cell death including both apoptosis and necrosis. So all cells that were positive for either of these two staining (AnV+, DAPI+ or AnV+DAPI+) were considered as cells undergoing cell death. In the figure cells that were negative for both AnnexinV and DAPI were considered as viable cells.

- Fig. 2D: the authors analyze phagocytosis of cells. The cell death analysis from the same experiments should be presented. How many AnV+/PI+,AnV+PI-, AnV-PI+ and AnV-PI- cells were given for the uptake with macrophages.

The starting materials (target cells – either neutrophils or cancer cells) were living cells directly from cell culture. We didn't perform induction of cell death. We have included data (Supplementary Fig. 2d, Supplementary Fig. 4a&4b, Supplementary Fig. 6a-6e) showing that majority of the cells (>95%) were viable (negative for either Annexin V or sytox Blue) before and after the assay unless they were engulfed by the macrophages.

- Fig 3: the authors demonstrate a comparative analysis of CRT expression in macrophages and neutrophils. Since the PI+ cells can be also positive for CRT, it is advised to exclude such cells from the analysis. In other words what is the % of viable cells in these experiments (Fig. 3A, 3D, 3E). These data should be included in the manuscript.

The starting materials (target cells – either neutrophils or cancer cells) were living cells directly from cell culture (Supplementary Fig. 2d, Supplementary Fig. 4a&4b, Supplementary Fig. 6a-6e). We didn't perform induction of cell death.

- What was the % of dead/viable cells in the phagocytosis experiments presented in the Fig. 5E, 5F, 5H, 5J). These data should be presented.

The starting materials (target cells – either neutrophils or cancer cells) were living cells directly from cell culture. We didn't perform induction of cell death. We have included data (Supplementary Fig. 2d, Supplementary Fig. 4a&4b, Supplementary Fig. 6a-6e) showing that majority of the cells (>95%) were alive (negative for either Annexin V or Sytox Blue) before and after the assay unless they were engulfed by the macrophages.

- Fig. 6: the authors should compare the cell death rate in vivo in the tumour. Since it is conceivable the cells are just dying more rapidly in 'Neu' condition.

We thank the reviewer for this suggestion. We would expect that because this was a transient treatment of neuraminidase the efficacy of the treatment would depend on the half-life of the desialyated proteins on the surface of cancer cells. While most the neuraminidase-treated cells were cleared by tumor macrophages after engrafting the mice, the surviving cells will restore their sialic acids decoration when they proliferate. So tumors in the neuraminidase-treated group of mice eventually grew up but were smaller because most of the engrafted cells were cleared during the initiation of the tumor.

To address the point whether neuraminidase treatment induces cell death and inhibits cell proliferation, we performed cell viability staining for control and neuraminidase treated cells and we showed there were no differences between these two groups. After a long-term cell culture, we observed no differences in cell growth. These data have been included in the manuscript (Supplementary Fig. 6a-6f).

- it is often very difficult to analyze phagocytosis of dead cells by flow cytometry because discrimination between attached cells and engulfed cells is quite problematic. For this pH sensitive probes for phagocytosis should be used. Please prove the you indeed measure phagocytosis and not cell attachment.

This is a very important point. We thank the reviewer for raising this concern. First of all, as we mentioned in the response to the questions above, we mainly focused on PrCR of living cells and majority of the target cells we used for our phagocytosis assay were living cells. The assay has been optimized and routinely performed in the laboratory (Jaiswal et al. 2009; Majeti et al. 2009; Chan et al. 2009; Chao et al. 2010; Chao et al. 2011; Willingham et al. 2012; Edris et al. 2012; Weiskopf et al. 2013; Feng et al. 2015; Weiskopf et al. 2016; Barkal et al. 2018; and many others). We have previously compared the phagocytosis results from microscopic-based and FACS-based assays and the results were consistent (Jaiswal et al. 2009; Majeti et al. 2009 etc. and many others). We have also previously confirmed that phagocytosis we observed with FACS-

based assays was indeed cellular engulfment instead of cell attachment by using PHrodo dyes.

We have now added additional data as suggested by the reviewer by staining cells with LysoTracker prior to phagocytosis assay and we showed the results were consistent that what we observed in FACS assay was indeed engulfment of target cells not just cell attachment (Supplementary Fig. 4c-4e).

- isotype control ABs should be included in all experiments (e.g. Figs 2A,2B, 2C; Fig3).

We thank the reviewer for point this out. We have included isotype staining or secondary control staining for all experiments.

Reviewer #3 (Remarks to the Author):

In this report the authors identify the endoplasmic reticulum protein calreticulin as a programmed cell removal (PrCR) signal. It was picked up as a candidate PrCR from a screen of activated macrophages from >300 genes that showed increased expression upon activation by toll like receptor ligands. Evidence suggests that CRT is transported to the cell surface of macrophages, and can then 'label' activated neutrophils by binding to asialo-glycoproteins, which are generated by the human neuraminidase, Neu4. The CRT then acts as an 'eat me' PrCR signal for macrophages.

While the proposal is interesting, and the data presented suggest how CRT can work as a factor in PrCR, it is not compelling. The rationale for production of CRT by macrophages and decoration of cells displaying 'eat me' signatures, in this case asialoglycoproteins, supports a consistent story, but each piece of evidence by itself is supportive in some cases and weak in others.

The topic of this paper [and our research] is not just normal cell removal or malignant cell removal. The topic is the mechanisms used by at least some macrophages to find the cells that bear the CRT target, to phagocytose them, whether malignant or aging, or otherwise pathologic. To us and hopefully to a journal like Nature Communications, the generality of this universal mechanism for homeostasis and removal of dangerous cells in vivo is extremely interesting and broadly important.

To reiterate, this paper was to explore the underlying mechanisms of PrCR. As we mentioned earlier, while PrCR is an important step for the clearance of dying cells, it's also involved in removing of living cells, which occurs during many biological and pathobiological processes such as aging and cancer. In this paper we investigated this mechanism by using tissue homeostasis, cancer and stem cell models. The mechanisms of cell clearance in these processes have not been elucidated so far. Consistent results in these important models strengthened our findings that calreticulin and its desialylated binding sites regulate immune cell recognition and promotes programmed cell removal. We are including this information to demonstrate that our findings have broad implications for potentially every normal and malignant cell. This pathway is central to our understanding of how cells are removed or evade removal in our bodies.

This paper illustrates the power of glycosciences in solving major problems in biomedical research. We believe studies of glycosciences whose roles in cancer and many pathophysiological processes are extremely important but far from being clearly elucidated, should be highly encouraged.

1. Of the many activated neutrophil genes that can be inferred to be relevant to PrCR (Fig. 1), it is not clear how CRT was selected for further study.

We apologize for not making this point clear. We have revised the manuscript accordingly.

The focus of this study is the mechanism of target cell recognition and phagocytosis by macrophages via PrCR, in cancer immunosurveillance and tissue homeostasis. We have *previously shown* that CRT is a critical effector mediating PrCR – blockade of CRT on macrophages inhibited PrCR while upregulating CRT on macrophages promoted PrCR of living cancer cells. In the current manuscript, we moved forward to further understand the molecular mechanisms of how CRT interact with its ligand on target cells to mediate PrCR. So CRT was picked for further study was based on our previous findings and our desire to understand the underlying mechanisms of PrCR of living cells.

We understand that traditionally it is believed that phagocytosis by macrophages is of dead cells, but our data starting in 1994 show that even without cell death, phagocytosis in PrCR is real, and can homeostatically regulate cell numbers. So in this study we used the MRP8-Bcl2 transgenic mouse model to show that when PCD was blocked by expressing the anti-apoptotic protein Bcl2 [and we showed that the Bcl2⁺ cells don't die during the experiment], PrCR still occurred. Both apoptosis and PrCR programs were activated in wild-type (WT) neutrophils to promote their removal in the peritoneum, while only PrCR was activated in the Bcl2-expressing neutrophils. Thus, we believe this is an excellent model for the purpose of studying PrCR due to its ability to distinguish PCD and PrCR pathways. And our RNAseq experiments comparing immature bone-marrow neutrophils to mature peritoneum-infiltrating neutrophils from WT and MRP8-Bcl2 mice identified potential pathways that are specifically involved in PrCR.

2. The correlation of CRT expression on macrophages and neutrophils over time following thioglycollate treatment is done without statistics (Fig. 2a-2c).

We have now included figures which incorporates multiple mice and statistical analysis for each time point for Fig2a, b, and c. Isotype controls are labeled in the figures. We consistently observed the same trend for CRT, CD47 and rCRT staining for pMN and macrophages, while we did observe some slight differences between different individual experiments including our initial graphs in MFI values, mostly due to variations between individual mice and experimental settings. We include in the manuscript a representative experiment from at least 3 mice for each time points.

3. Data suggesting that CRT recognizes asialoglycoproteins relies heavily on binding to a glycan array result showing that it selectively binds to asialo-orosomucoid (ASOR), which is a serum glycoprotein, alpha 1 acid glycoprotein. CRT has been run on the much larger glycan arrays of the Consortium for Functional Glycomics by several groups (functionalglycomics.org), and results were inconclusive (no binding) or no binding to galactose terminated glycans, including tri-antennary N-linked glycans terminated with galactose.

We appreciate the reviewer for this criticism because it points out the strength of our approach. The functional glycomics consortium is a great resource, however the results that from these arrays can provide useful information under assumptions that we did not want to make. For examples, there are only three experiments that we can find that have been submitted to the consortium that have used full-length recombinant human CRT whereas the other entries are different isoforms or truncated forms of CRT. Additionally, there is no indication how the protein was generated, for example we express our CRT in mammalian cells. We cannot explain the observations that are deposited into the consortium but try to include controls for glycan binding and different strategies to detect CRT binding.

The binding of CRT to ASOR has been repeated multiple times in this study, with purified CRT with different tags including His-, mouse Fc and human Fc, and we received consistent results from these experiments. We think the results could be affected by the concentrations of purified proteins used for probing (we showed that stronger binding was observed when 0.25ug/ml of mammalian CRT recombinant proteins were used than the condition of 0.05ug/ml), origin of glycans used for glycan arrays, as well as the origin of recombinant proteins (we purified recombinant CRT proteins using mammalian cells instead of yeast or bacteria cells so the posttranslational modifications could be preserved, which are likely to affect the binding between CRT to its ligand in mammalian cells).

In addition, we have examined CRT binding to cell surface glycan molecules, by using WT and mutant CHO cell lines that specifically express Tri/m-II and its derivatives. Consistently, stronger binding of recombinant CRT was observed on CHO lines defective in the CMP-sialic acid transporter (Lec2) or UDP-GlcNAc 2 epimerase (Lec3) that expressed Tri/m-II, as compared to parental line, or the line defective in UDP-Gal translocase that expressed Tri/m-II capped by sialic acids or missing mannose on the outermost layer (Supplementary Fig. 3b).

Lastly, it's possible that Tri/m-II is one of the glycan-binding partners of CRT. We are currently working on screening for additional glycans expressed on tumor cells that interact with CRT, as well as identifying glycosylated proteins bearing glycan structures that bind to CRT. These questions and further characterizations will be conducted in future publications.

4. Use of exogenous neuraminidases to show changes in lectin binding and phagocytosis in Fig. 5 are correlative, but exogenous bacterial neuraminidases dramatically desialylate surface glycoproteins and create artifacts that may or may not be connected to the biology being studied. There is no direct connection to Neu4 or CRT biology.

The idea of asialoglycans as “eat me” signals on target cells that can be detected by macrophage-originating CRT to enable PrCR were carefully examined by multiple models and methods in our study. In addition to neuraminidase treatment, we have

showed a series of evidences for the relevance of this pathway to cancer, including 1) we examined the expression of identified asialoglycan antigens on multiple types of cancer cells (Figures 4e, 4f, 5g, S3b, S3d, etc.); 2) we showed a direct correlation of key regulators of the identified pathways (NEU1-4, ST3GAL1, ST6GAL1 etc.) to cancer patient survival (Figures 6c-6f); 3) we indeed showed with genetic approaches that knockout of key regulators of the identified pathways substantially reduced cancer cells' susceptibility to PrCR (Figures 5g and 5h); 4) we showed with a combination of *in vitro* and *in vivo* data that neuraminidase treatment generated CRT-binding asialoglycan antigens on multiple types of cancer cells and such treatment led to an enhanced PrCR of these cells and blockade of their engraftment in NSG mice (Figures 5e, 5f, 6a, 6b, S5a-5c); 5) we examined the expression of CRT and CRT-binding asialoglycan antigens on AML patient samples (Figures 6g and 6h); 6) we showed independent binding to the asialoglycan specific lectin, PHA-L, and showed by co-capping that calreticulin and PHA-L bind to the same molecular species on the cell surface, namely asialoglycoprotein; 7) we have showed using CHO cell model that CRT selectively bind to Tri/m-II structures than Tri/m-II glyans that were capped by sialic acids. A combination of these results consistently indicated the expression and function of CRT-binding asialoglycan antigens on cancer cells and their roles in PrCR. The use of bacterial neuraminidase in some experiments was not to implicate it in physiology, but to have an independent method to implicate asialoglycans as the CRT target.

5. Use of oseltamivir as a neuraminidase inhibitor in Fig. 5g&h is puzzling, since this is an influenza virus inhibitor that has been shown not to inhibit human Neu1, 2, 3, or 4.

We thank the reviewer for bringing up this point. We agree with the reviewer that many studies have demonstrated that oseltamivir inhibits with preference influenza neuraminidase but many of the previous studies also showed mammalian neuraminidase activity can be inhibited with oseltamivir (PMID 19430901, 20347965, 25850034 etc.). While it's likely oseltamivir might have differential preference towards mammalian Neu 1-4, we believe the most direct experiment to address this is to examine the binding affinity of oseltamivir to purified human Neu1, 2, 3, 4 proteins with biochemical assays. Before this can be clearly addressed by us or others, in an independent project in the future, we removed this figure from the current manuscript.

6. Use of PHA to detect binding sites for CRT in neuraminidase treated HL60 cells (Fig. 6a&b) is overstating the conclusion that CRT has the same specificity as PHA. This has not been shown.

We have a series of studies showing PHA-L can be used for detecting CRT binding sites on target cells: 1) PHA-L mimics CRT binding kinetics on neutrophils and macrophages in the thioglycollate-treated Bcl2 transgenic mouse model; 2) recombinant CRT proteins and PHA-L colocalized on the cell surface of cancer cells; 3) PHA-L labels asialoglycans after neuraminidase treatment on cancer cells, similarly as CRT

recombinant proteins; 4) PHA-L has been demonstrated to specifically bind with Tri/m-II dominant structures previously.

However, we agree with the reviewer that it is possible that PHA-L may potentially label asialoglycans other than Tri/m-II. We will develop monoclonal antibody targeting CRT-binding sites for the future studies. As the reviewer has pointed out, we don't want to overstate our findings. We have used both recombinant CRT proteins and PHA-L to examine CRT binding to cancer cells after neuraminidase treatment. We have revised the manuscript accordingly so it would more accurately reflect our findings.

7. While the correlation of survival with decreased expression of sialyltransferases or increased expression of Neu4 adds biological significance, the association of high sialic acid and poor prognosis has been around for decades, and there is no clear tie to the CRT/PrCR mechanism for this correlation.

In this study, we demonstrated that PrCR is a general mechanism for the clearance of aged, dysfunctional and malignant cells. We showed that the interaction between CRT and asialoglycans on target cells is critical for the recognition and phagocytosis of these cells in PrCR. Despite the importance of glycosciences, the resources for such important studies are very limited, especially clinically relevant database. Although we don't have clinical data showing direct correlation between CRT binding sites on cancer cells and cancer patients survival, due to the very limited resources of studies investigating cancer cell glycosylation and clinical outcomes, we examined neuraminidase/sialyltransferase pathways that directly regulate unmasking and masking of CRT binding sites and we showed a strong correlation between the availability of these sites and patient survival. In addition, we carefully examined CRT/PrCR mechanisms in multiple models including tissue homeostasis and stem cells. Consistent results were achieved from these studies. We combined a series of *in vitro* and *in vivo* studies as described in response to question 4, showing the important role of CRT-asialoglycan axis in PrCR in multiple biological and pathophysiological processes.

In the following up studies, we will further elucidate the underlying mechanisms of CRT-glycan interactions in tumor immunosurveillance. While altered glycosylation has been regard as a hallmark of cancer, the role of glycosylation in cancer cell immune evasion has remained unclear. We believe that such studies should be highly encouraged.

Reviewers' comments:

Reviewer #1 (Remarks to the Author):

In my assessment the authors have satisfactorily answered the concerns and the MS is now acceptable for publication.

Reviewer #2 (Remarks to the Author):

In general, the manuscript has been improved but still some of the previous comments have been not fully addressed by the authors.

An important question remains in regard to the Figs. 2A,B,C. The fig. is a reflection of the wild type neutrophils. However, from the Fig. 1 it is clear that wild type neutrophils undergo cell death, and this is not a surprise that they do become CRT positive. Since the authors claim that viable BCL2 neutrophils are engulfed, it is strongly advised to provide the similar data sets on these BCL2 expressing neutrophils which supposed to be viable and it is claimed that they are engulfed in CRT-dependent manner. These data are important in order to generalize the fact that viable neutrophils are engulfed.

The comment on the Fig 2D is not addressed by the authors. The authors refer to Suppl. Fig. 2D- those are obviously macrophages. The concern still remains.

The question was/is to demonstrate cell death rate for WT and BCL2-MRP8 neutrophils which have been used exactly in the phagocytosis assay. So the question still remains on the Fig. 2D: what was the rate of cell death and viable cells in WT and BCL2-MRP8 neutrophils before setting up phagocytosis assay? This is one of the important figures to confirm the main conclusions of the manuscript in regard to the clearance of viable neutrophils.

Fig. 3A, E, G and 4C,D: these are the wild type cells (i.e dying) but since the main focus of the first part of the study is engulfment of viable neutrophils it is necessary to show the similar data sets for BCL-MRP8 neutrophils and not only for wild type cells, which undergo cell death. The cell death markers (e.g. Sytox/AnnexinV) have to be also included.

The authors do not provide evidence that CRT could be bound to viable neutrophils (i.e., BCL2-MRP8 neutrophils). This is a crucial question for the first part of the manuscript and therefore it is needed to be addressed by the authors.

Appropriate statistical methods should be used e.g. a two-way ANOVA test. t student test is used just to compare two groups and not suited for this analysis . If there are more than two groups other tests have to be used.

Minor comments

In general, the results in the manuscript needed still to be better explained. Some of the figures are not thoroughly enough explained. Hereby I provide some comments which will help to improve the readability of the manuscript:

- A consist definition of programmed cell removal (PrCR) has to be provided throughout the whole manuscript. For example, in the abstract it is defined as: "Macrophage-mediated programmed cell removal (PrCR) is a process critical for the clearance of unwanted (damaged, dysfunctional, aged or harmful) cells."

While in the first sentence in the introduction it is defined as "We have termed the process of viable cell clearance through phagocytosis by macrophages as "programmed cell removal" (PrCR), ..."

The question is whether programmed cell removal (PrCR) is clearance of viable cells only or all cells i.e. viable and dead/damaged cells?

- Fig 2D: it is not explained in the text and I do not see any difference in CD47 MFI. What is the reason to show this figure? Either it has to be explained and statistical analysis included or it has to be omitted.

- On page 7 it is stated: "Interestingly, as the cell surface levels of CRT markedly decreased on the peritoneal macrophages the cell surface levels increased on the neutrophils over time which correlated with the decreased levels of neutrophils observed in the peritoneum."

From the 2A we could not conclude that the surface CRT levels in macrophages are increased. From the presented figure one can see that it is unchanged. Such conclusions can be drawn only when they are supported by statistical analysis (in regard to the macrophages). Please rephrase accordingly.

- From all figure legends it is not clear to which groups conditions marked by "stars" are compared. Please add appropriate explanations to the figure legend. This is true for all the figure legends including supplementary; they have to be adapted.

- The authors have to better explain why anti-SIRP1 and anti-CD47 have been used in the figures 5E, F, H.

Reviewer #3 (Remarks to the Author):

In this revised manuscript, the authors satisfactorily addressed the concerns of this referee.

Reviewer #2 (Remarks to the Author):

In general, the manuscript has been improved but still some of the previous comments have been not fully addressed by the authors.

An important question remains in regard to the Figs. 2A,B,C. The fig. is a reflection of the wild type neutrophils. However, from the Fig. 1 it is clear that wild type neutrophils undergo cell death, and this is not a surprise that they do become CRT positive. Since the authors claim that viable BCL2 neutrophils are engulfed, it is strongly advised to provide the similar data sets on these BCL2 expressing neutrophils which supposed to be viable and it is claimed that they are engulfed in CRT-dependent manner. These data are important in order to generalize the fact that viable neutrophils are engulfed.

As shown in the figure1 and also our previous studies (Lagasse, E, and IL Weissman (1994). *bcl-2 inhibits apoptosis of neutrophils but not their engulfment by macrophages. J Exp Med.* **179**(3): 1047-52.

http://www.ncbi.nlm.nih.gov/entrez/query.fcgi?cmd=Retrieve&db=PubMed&dopt=Citation&list_uids=8113673), while Bcl2 neutrophils are resistant to cell death they share the same programmed removal pathway with WT neutrophils. Therefore, in most of the experiments carried out in this study for this manuscript, Bcl2 mice are used as the source of peritoneal macrophages and neutrophils, because the main focus of our study is PrCR of viable cells.

The data in Fig2a, b, c are all cells from the MRP8-Bcl2 mice. We apologize for any confusions and we have updated the figure legends to make them clearer.

The comment on the Fig 2D is not addressed by the authors. The authors refer to Suppl. Fig. 2D- those are obviously macrophages. The concern still remains. The question was/is to demonstrate cell death rate for WT and BCL2-MRP8 neutrophils which have been used exactly in the phagocytosis assay. So the question still remains on the Fig. 2D: what was the rate of cell death and viable cells in WT and BCL2-MRP8 neutrophils before setting up phagocytosis assay? This is one of the important figures to confirm the main conclusions of the manuscript in regard to the clearance of viable neutrophils.

For the experiment in Fig2d, cells were collected 4hrs after thioglycollate injection. We know from our data as represented in Fig1b (viability measured by AnnexinV and Dapi), up to 24hrs in in vitro culture condition, around 80% of the neutrophils are viable and therefore no significant difference between WT and Bcl2 neutrophils during this time interval. Therefore, the majority of neutrophils were viable before conducting the phagocytosis assay in Fig2d.

We have updated the manuscript and figure legends to make it clearer.

Fig. 3A, E, G and 4C,D: these are the wild type cells (i.e dying) but since the main focus

of the first part of the study is engulfment of viable neutrophils it is necessary to show the similar data sets for BCL-MRP8 neutrophils and not only for wild type cells, which undergo cell death. The cell death markers (e.g. Sytox/AnnexinV) have to be also included.

In most of the experiments we showed in this manuscript, Bcl2 mice are used as the source of peritoneal macrophages and neutrophils, because the main focus of our study is PrCR of viable cells.

The data in Fig2a, b, c are all cells from the MRP8-Bcl2 mice.

We apologize for any confusions and we have updated the figure legends to make them clearer.

The authors do not provide evidence that CRT could be bound to viable neutrophils (i.e., BCL2-MRP8 neutrophils). This is a crucial question for the first part of the manuscript and therefore it is needed to be addressed by the authors.

These data are already included in Fig2c, Fig4c and 4d where recombinant CRT proteins were used to incubate with viable neutrophils and we observed binding of CRT proteins to these cells.

Appropriate statistical methods should be used e.g. a two-way ANOVA test. t student test is used just to compare two groups and not suited for this analysis. If there are more than two groups other tests have to be used.

All the comparison in the manuscript are between two groups. In Fig2a, b, c, we were comparing difference of CRT, CD47 and rCRT on peritoneal macrophages and neutrophils. We conducted paired t test for our statistical grouped analysis.

We apologize for any confusions and we have updated the figure legends to make them clearer.

Minor comments

In general, the results in the manuscript needed still to be better explained. Some of the figures are not thoroughly enough explained. Hereby I provide some comments which will help to improve the readability of the manuscript:

- A consist definition of programmed cell removal (PrCR) has to be provided throughout the whole manuscript. For example, in the abstract it is defined as: "Macrophage-mediated programmed cell removal (PrCR) is a process critical for the clearance of unwanted (damaged, dysfunctional, aged or harmful) cells."

While in the first sentence in the introduction it is defined as "We have termed the process of viable cell clearance through phagocytosis by macrophages as "programmed cell removal" (PrCR), ..."

The question is whether programmed cell removal (PrCR) is clearance of viable cells only or all cells i.e. viable and dead/damaged cells?

We appreciate the reviewer's guidance and suggestions for the purpose to making the manuscript clearer. In response to this specific point:

Traditionally PrCR was thought of as the last step of apoptosis to clear the apoptotic cells. But in many conditions, as documented in many previous studies including ours, PrCR could also occur independently of apoptosis, namely in addition to the clearance of apoptotic cells, PrCR also leads to direct clearance of viable cells, which is also the main focus of this study. Therefore, PrCR is a pathway that can promote the clearance of all cells (viable and or dead/damaged) by macrophages.

We have revised the introduction part of the paper accordingly.

- Fig 2D: it is not explained in the text and I do not see any difference in CD47 MFI. What is the reason to show this figure? Either it has to be explained and statistical analysis included or it has to be omitted.

We thank the reviewer for bringing this very important observation into discussion. The efficacy of PrCR is determined by the balance between the recognition of pro-phagocytic "eat me" signals (CRT) by macrophages and the inhibition of macrophages via the activation of anti-phagocytic "don't eat me" pathways by target cells. CD47 is the major "don't eat me" signal expressed so we included both CD47 and CRT in our analysis to understand their roles in this process. In addition, CD47 staining is also considered as a good control for CRT staining.

In this induced peritonitis model we observed no significant change of CD47. We have updated this information in manuscript and figure legends.

- On page 7 it is stated: "Interestingly, as the cell surface levels of CRT markedly decreased on the peritoneal macrophages the cell surface levels increased on the neutrophils over time which correlated with the decreased levels of neutrophils observed in the peritoneum."

From the 2A we could not conclude that the surface CRT levels in macrophages are increased. From the presented figure one can see that it is unchanged. Such conclusions can be drawn only when they are supported by statistical analysis (in regard to the macrophages). Please re-phrase accordingly.

We thank the reviewer for this suggestion. We now revised this sentence to make it clearer:

"Interestingly, the cell surface levels increased on the neutrophils over time which correlated with the decreased levels of neutrophils observed in the peritoneum."

- From all figure legends it is not clear to which groups conditions marked by “stars” are compared. Please add appropriate explanations to the figure legend. This is true for all the figure legends including supplementary; they have to be adapted.

We apologize for any confusions and we have updated the figure legends to make them clearer.

- The authors have to better explain why anti-SIRP1 and anti-CD47 have been used in the figures 5E, F, H.

CD47 interact with Sirpa to transmit a negative signal to eventually inhibit phagocytosis. Therefore, antibodies blocking CD47 or Sirpa would both lead to induction of phagocytosis by promoting the “eat me” pathway to prevail, as we also documented in our previous studies. We included this approach to isolate PrCR by two different approaches to determine the effects of neuraminidase treatment in the PrCR pathway.

REVIEWERS' COMMENTS:

Reviewer #2 (Remarks to the Author):

The authors have significantly improved the manuscript and addressed all comments. These results are very interesting and intriguing. The manuscript can be published in the journal.